# ELASTIC OPTIMAL TRANSPORT: THEORY, APPLICATION, AND EMPIRICAL EVALUATION

**Pei Yang, Yuhang Zhuang**
Department of Computer Science
South China University of Technology
Guangzhou, China
yangpei@scut.edu.cn
202421044690@mail.scut.edu.cn

**Qi Tan**
Department of Computer Science
South China Normal University
Guangzhou, China
tanqi@scnu.edu.cn

## ABSTRACT

The classical optimal transport such as Kantorovich's optimal transport and partial optimal transport could be too restrictive in applications due to the full-mass or fixed-mass preservation constraints. To remedy this limitation, we propose elastic optimal transport (ELOT) which is distinctive from the classical optimal transport in its ability of adaptive-mass preserving. It aims to answer the problem of how to transport the probability mass adaptively between probability distributions, which is a fundamental topic in various areas of artificial intelligence. The strength of elastic optimal transport is its capability to transport adaptive-mass in the light of the geometry structure of the problem itself. As an application example in machine learning, we apply elastic optimal transport to both unsupervised domain adaptation and partial domain adaptation tasks. It adaptively transports masses from source domain to target domain by taking domain shift into consideration and respecting the ubiquity of noises or outliers in the data, in order to improve the generalization performance. The experiment results on the benchmarks show that ELOT significantly outperforms the state-of-the-art methods. As a powerful distribution matching tool, elastic optimal transport might be of interests to the broad areas such as artificial intelligence, healthcare, physics, operations research, urban science, etc. The source code is available in the supplementary material.

## 1 INTRODUCTION

The optimal transport (OT) problem first came up in Monge's seminal work (Monge, 1781). It aims to find an optimal way to move a pile of sand into a hole, assuming the pile and the hole must have the same volume. OT is formulated as the mathematical problem of comparing two probability distributions, which is a fundamental problem in many areas. Many researchers in different areas found that optimal transport was strongly linked to their subjects, and helped expand the optimal transport foundations (Villani, 2003). Recent years have witnessed another revolution in the spread of OT, thanks to the emergence of approximate algorithms that can solve large-scale problems (Cuturi, 2013). As a consequence, OT is being increasingly used to unlock various problems in artificial intelligence, statistics, bioinformatics, economics, logistics, physics, etc.

However, the classical optimal transport has its limitations, which motivates us to develop the novel formulation of OT. Kantorovitch (1942) moved away from the idea that mass transportation should be deterministic to consider instead a probabilistic transport, which allows mass splitting from a source toward several targets. But a major bottleneck of OT in its traditional formulation is that it requires the two distributions to have the same total probability mass and all probability mass has to be transported. This is too restrictive in many applications because full mass preservation is likely to fit noise and outliers, or undesired pairs, and prevent any form of partial matching. There have been two main directions, i.e., partial optimal transport and unbalanced optimal transport, to attempt to remove the constraints of full-mass preservation. Caffarelli & McCann (2010) and Figalli (2010) proposed partial optimal transport to preserve the fixed amount of mass instead the full mass. Liero et al. (2017) developed unbalanced optimal transport which replaced the 'hard' marginal constraints

of OT by 'soft' penalties using some divergence. Both partial optimal transport and unbalanced optimal transport provide great flexibility to model partial matching. However, they lack the ability of adaptive-mass transport. For partial optimal transport, it is challenging to determine the fixed budget of mass to transport. For unbalanced optimal transport, it is usually unknown to what extent the 'soft' penalties should be imposed on the marginal constraints.

Therefore, a natural question is: could it adaptively transport probability masses among probability distributions? This motivates us to develop the *elastic optimal transport*, which is able to adaptively determine the transported masses in the light of the intrinsic structure of the problem itself. The distinctive advantage of elastic optimal transport against its classical counterparts lies in the ability of adaptive-mass preserving. The elastic optimal transport is of great interests to the broad areas such as machine learning, biomedical, computational physics, operations research, urban science, etc. For example, optimal transport plays a crucial role in a wide variety of machine learning applications, such as generative adversarial networks (Arjovsky et al., 2017), computer vision (Solomon et al., 2016), natural language processing (Xu et al., 2021), clustering (Ho et al., 2017), graph matching (Zhang & Lauw, 2024; Sun et al., 2024), semi-supervised learning (Chapel et al., 2020), and domain adaptation (Courty et al., 2017b). The essential problem in these applications is how to compare two probability distributions (e.g., align the AI-generated images with the natural images). Due to the ubiquity of noises, outliers, and divergences existing in the real-world data, the optimal transport with adaptive-mass preserving is preferred in such applications to alleviate the distortion of OT mappings. The main contributions of the paper are highlighted as follows.

- We propose a novel formulation of optimal transport. The elastic optimal transport is distinctive from classical optimal transport in the aspects of adaptive-mass preserving and mixed-sign ground cost matrix. Its strength is that the transported masses are adaptively determined by the geometry structure of the problem, providing a powerful tool for adaptive distribution matching.

- The theoretical analysis sheds light on the mass transport mechanism of ELOT. Furthermore, we derive an equivalent formulation for elastic optimal transport, which can be solved by the off-the-shelf OT algorithms.

- We propose a novel model for domain adaptation based on elastic optimal transport. It transports the masses from the source domain to the target domain adaptively, taking full consideration of the domain shift. The experiment results on the benchmarks show that ELOT outperforms the state-of-the-art approaches in a variety of unsupervised domain adaptation and partial domain adaptation tasks.

## 2  ELASTIC OPTIMAL TRANSPORT

We introduce the classical optimal transport, and then propose the elastic optimal transport approach.

### 2.1  PRELIMINARY

Without loss of generality, we focus on discrete optimal transport in this paper. Assume that we are given a pile of sand, and a hole that we have to completely fill up with the sand. Let $\mu$ and $\nu$ be the probability measures of the pile and the hole respectively. The probability measure $\mu$ with the probability masses $\{p_i\}_{i=1}^n$ on the locations $\{x_i\}_{i=1}^n$ is formulated as $\mu = \sum_{i=1}^n p_i \delta_{x_i}$ where $\delta_{x_i}$ is the Dirac function located at $x_i$, and $p_i$ belongs to the probability simplex, i.e. $\sum_{i=1}^n p_i = 1$. Likewise, the probability measure $\nu$ with the probability masses $\{q_j\}_{j=1}^m$ on the locations $\{z_j\}_{j=1}^m$ is formulated as $\nu = \sum_{j=1}^m q_j \delta_{z_j}$. The transport plans are formulated by the probability measures $\gamma$, where $\gamma_{ij}$ measures the amount of mass transported from location $x_i$ to location $z_j$. The bounded ground cost $\mathcal{C}_{ij}$ tells how much it costs to transport one unit of mass from $x_i$ to $z_j$. We denote $\mathbb{R}_+^{n \times m}$ and $\mathbb{R}_\pm^{n \times m}$ as the non-negative and mixed-sign matrix, respectively. The $d$-dimensional vector of ones is denoted by $\mathbf{1}_d$.

**Kantorovich's Optimal Transport.** In the discrete setting, Kantorovich's optimal transport problem (Kantorovitch, 1942) is formulated as:

$$\min_{\gamma \in \Pi(\mu,\nu)} <\gamma, \mathcal{C}> \tag{1}$$

where $\mathcal{C} \in \mathbb{R}_+^{n \times m}$ and

$$\Pi(\mu, \nu) = \{\gamma \in \mathbb{R}_+^{n \times m} \mid \gamma \mathbf{1}_m = \mu, \gamma^T \mathbf{1}_n = \nu\}. \tag{2}$$

**Partial Optimal Transport.** Partial optimal transport (Caffarelli & McCann, 2010; Figalli, 2010) relaxes the full-mass constraints in Kantorovich's problem and preserves the fixed amount of mass:

$$\min_{\gamma \in \Pi_p(\mu, \nu)} <\gamma, \mathcal{C}> \tag{3}$$

where $\mathcal{C} \in \mathbb{R}_+^{n \times m}$ and

$$\Pi_p(\mu, \nu) = \{\gamma \in \mathbb{R}_+^{n \times m} \mid \gamma \mathbf{1}_m \leq \mu, \gamma^T \mathbf{1}_n \leq \nu, \mathbf{1}_n^T \gamma \mathbf{1}_m = s\} \tag{4}$$

where $s(0 \leq s \leq min(\|\mu\|_1, \|\nu\|_1))$ is the fixed budget of mass, which is user-defined. $\|\cdot\|_1$ is the $l_1$ vector norm.

**Unbalanced Optimal Transport.** Unbalanced optimal transport (Liero et al., 2017) relaxes the 'hard' marginal constraints in Kantorovich's problem with the 'soft' penalties by using some divergence $\mathcal{D}_\phi$:

$$\min_{\gamma \in \mathbb{R}_+^{n \times m}} <\gamma, \mathcal{C}> +\tau_1 \mathcal{D}_\phi(\gamma \mathbf{1}_m \| \mu) + \tau_2 \mathcal{D}_\phi(\gamma^T \mathbf{1}_n \| \nu) \tag{5}$$

where $\mathcal{C} \in \mathbb{R}_+^{n \times m}$. The trade-off coefficients $\tau_1$ and $\tau_2$ are used to adjust the weights of the marginal divergences.

## 2.2 ELASTIC OPTIMAL TRANSPORT

We propose the elastic optimal transport in order to overcome the limitations of the classical optimal transport, which are discussed in Section 1. In the discrete setting, elastic optimal transport is formulated as:

$$\mathcal{W}(\mu, \nu) = \min_{\gamma \in \Pi_e(\mu, \nu)} <\gamma, \mathcal{C}> \tag{6}$$

where $\mathcal{C} \in \mathbb{R}_\pm^{n \times m}$ and

$$\Pi_e(\mu, \nu) = \{\gamma \in \mathbb{R}_+^{n \times m} \mid \gamma \mathbf{1}_m \leq \mu, \gamma^T \mathbf{1}_n \leq \nu\}. \tag{7}$$

The distinctive characteristics of elastic optimal transport are twofold: adaptive-mass preserving and mixed-sign ground cost matrix. First, the partial masses are adaptively transported. There are no full-mass or fixed-mass preserving constraint as done in Kantorovich's optimal transport and partial optimal transport, respectively. Second, the ground cost matrix $\mathcal{C}$ is a mixed-sign matrix which contains positive and negative entries. The classical optimal transport usually assumes that the ground cost is non-negative (Villani, 2003). However, in many scenarios, it is natural to allow for negative costs. Consider the example related to $CO_2$ abatement from economics and operations research context (Levihn, 2016). Many investment options both increase productivity and reduce $CO_2$ emissions, leading to financial returns. Therefore, the costs for these options should be negative. In contrast, the costs for those investment options with financial expenditure should be positive.

In the seminal work of partial optimal transport, Caffarelli & McCann (2010) (equations 1.7 and 1.8) introduced the Lagrange multiplier $\lambda \geq 0$ and replaced the cost $c(x, y)$ with $c(x, y) - \lambda$, allowing for negative cost. Then the mass $s$ can be attained by selecting the appropriate value of $\lambda$. The drawback of Caffarelli & McCann (2010) is that it needs to specify the fixed budget of mass $s$. In contrast, ELOT is able to automatically find the optimal mass to be transferred. We overcome the drawback of Caffarelli & McCann (2010) and rely on the combination of mixed-sign cost and the marginal inequality constraints to achieve the goal of adaptive-mass preserving.

In summary, the advantage of elastic optimal transport lies in the self-adaptive preserving of mass. The elastic optimal transport is capable of preserving the suitable masses in accordance with the native geometry of the problem. It provides an elegant solution for adaptive matching. Unlike partial optimal transport or unbalanced optimal transport, it does not need to specify the fixed budget of mass or the softness of marginal constraints, which are challenging in essence.

Using the technique to convert the inequality into an equation by adding the slack variable in linear programming (Matoušek & Gärtner, 2007), we solve the elastic optimal transport problem by adding

slack variables and extending the cost matrix. Define the augmented cost matrix and the marginal vectors as

$$\bar{\mathcal{C}} = \begin{bmatrix} \mathcal{C} & \sigma \mathbf{1}_m \\ \sigma \mathbf{1}_n^T & 2\sigma \end{bmatrix}, \quad \bar{\mu} = \begin{bmatrix} \mu \\ \|\nu\|_1 \end{bmatrix}, \quad \bar{\nu} = \begin{bmatrix} \nu \\ \|\mu\|_1 \end{bmatrix} \tag{8}$$

where $\sigma \geq 0$ is a fixed non-negative scalar. The elastic optimal transport with inequality constraints can be reformulated as the following optimal transport problem with equality constrains

$$\mathcal{W}(\bar{\mu}, \bar{\nu}) = \min_{\gamma \in \Pi(\bar{\mu}, \bar{\nu})} < \gamma, \bar{\mathcal{C}} > \tag{9}$$

where

$$\Pi(\bar{\mu}, \bar{\nu}) = \left\{ \gamma \in \mathbb{R}_+^{(n+1) \times (m+1)} \mid \gamma \mathbf{1}_{(m+1)} = \bar{\mu}, \gamma^T \mathbf{1}_{(n+1)} = \bar{\nu} \right\}. \tag{10}$$

Theorem 1 shows the equivalence between the elastic optimal transport problem defined in Equation 6 and the reformulation defined in Equation 9, which allows using the off-the-shelf optimal transport algorithms to solve the elastic optimal transport problem.

**Theorem 1 (Optimal Transport Solution)** *Let $\boldsymbol{\gamma}^*$ (or $\bar{\boldsymbol{\gamma}}^*$) and $\mathcal{W}(\mu, \nu)$ (or $\mathcal{W}(\bar{\mu}, \bar{\nu})$) be the optimal transport plan and the optimal transport distance for the elastic optimal transport problem (or the reformulation). Assume $\bar{\boldsymbol{\gamma}}_o^*$ is the matrix $\bar{\boldsymbol{\gamma}}^*$ with the last row and column removed.*
*i) **Optimal Transport Plan.** The optimal transport plan $\boldsymbol{\gamma}^*$ is equivalent to $\bar{\boldsymbol{\gamma}}_o^*$:*

$$\bar{\boldsymbol{\gamma}}_o^* = \arg\min_{\gamma \in \Pi_e(\mu, \nu)} < \gamma, \mathcal{C} > = \boldsymbol{\gamma}^*. \tag{11}$$

*ii) **Optimal Transport Distance.** The difference of $\mathcal{W}(\mu, \nu)$ and $\mathcal{W}(\bar{\mu}, \bar{\nu})$ is a constant term which can be ignored in optimization (i.e., the optimization is not affected by the specific value of $\sigma$):*

$$\mathcal{W}(\bar{\mu}, \bar{\nu}) - \mathcal{W}(\mu, \nu) = \sigma \Big( \|\mu\|_1 + \|\nu\|_1 \Big). \tag{12}$$

The proof of Theorem 1 is in Appendix A.1.1. Also, Theorem 1 indicates that the optimal transport plan of ELOT keeps unchanged no matter what is the value of $\sigma$. So we just set $\sigma = 1$ in the experiments. The algorithm complexity of ELOT is slightly different from classical optimal transport with the scale of problem-solving changing from $n \times m$ to $(n+1) \times (m+1)$, where the extra one corresponds to the augmented dimension.

**Entropic Elastic Optimal Transport.** The entropy-regularized optimal transport (Cuturi, 2013) has the advantages that it defines a strongly convex problem which can be solved efficiently. Likewise, one may add the entropy-regularized term $\epsilon \mathcal{H}(\gamma) = -\epsilon \sum_{i,j} \gamma_{i,j} \log \gamma_{i,j}$ to the elastic optimal transport defined in Equation 6, where $\epsilon$ is the entropic coefficient. The entropy-regularized term encourages the sparsity of the transport plan, and allows using the Sinkhorn-Knopp algorithm (Cuturi, 2013) for efficient computation.

Note that either ELOT or classical optimal transport does not guarantee a unique solution, which stems from the interplay between its linear objective function and the convex polyhedral feasible region. Therefore, both the original ELOT and the reformulated problem could have multiple optima. However, the uniqueness of solution to the entropy-regularized ELOT and its reformulated counterpart is guaranteed because the entropy-regularized objective function is strictly convex.

## 2.3 MASS TRANSPORT MECHANISM

We cast the elastic optimal transport problem as a constrained worst transport problem, which provides insights into the mass transport mechanism of the elastic optimal transport problem.
Let's detach the cost matrix into the positive part $\mathcal{C}^+$ and the negative part $\mathcal{C}^-$, i.e., $\mathcal{C} = \mathcal{C}^+ + \mathcal{C}^-$, where

$$\mathcal{C}_{ij}^+ = \begin{cases} \mathcal{C}_{ij}, & \text{if } \mathcal{C}_{ij} > 0; \\ 0, & \text{otherwise.} \end{cases} \quad \mathcal{C}_{ij}^- = \begin{cases} \mathcal{C}_{ij}, & \text{if } \mathcal{C}_{ij} < 0; \\ 0, & \text{otherwise.} \end{cases}$$

We stipulate that the transport mass is zero (i.e., $\gamma_{ij} = 0$) when the cost is zero (i.e., $C_{ij} = 0$) to avoid ambiguity. The elastic optimal transport problem defined in Equation 6 is rewritten as

$$\min_{\gamma_{ij} \in \Gamma_e(\mu, \nu)} \sum_{i,j} \mathcal{C}_{ij} \gamma_{ij} \tag{13}$$

where

$$\Gamma_e(\mu, \nu) = \left\{ \gamma_{ij} \big| \sum_j \gamma_{ij} \leq \mu_i, \sum_i \gamma_{ij} \leq \nu_j \right\} \tag{14}$$

where $\mathcal{C}_{ij}$ could be positive or negative and $\gamma_{ij} (\geq 0)$ satisfies both left and right marginal inequality constraints for all admissible $i, j$. Then, we define the constrained worst transport problem

$$\max_{\gamma_{ij} \in \Gamma_{\mathcal{C}}(\mu, \nu)} \sum_{i,j} -\mathcal{C}_{ij}^- \gamma_{ij} \tag{15}$$

where

$$\Gamma_{\mathcal{C}}(\mu, \nu) = \left\{ \gamma_{ij} \big| \gamma_{ij} = 0, \text{ if } \mathcal{C}_{ij} = 0; \quad \sum_j \gamma_{ij} = \mu_i \text{ } \textbf{or} \text{ } \sum_i \gamma_{ij} = \nu_j, \text{ if } \mathcal{C}_{ij} < 0. \right\} \tag{16}$$

where $-\mathcal{C}_{ij}^-$ is non-negative and $\gamma_{ij} (\geq 0)$ satisfies the conditions specified in Equation 16 for all admissible $i, j$.

We prove the equivalence between elastic optimal transport and its corresponding constrained worst transport problem.

**Theorem 2 (Mass Transport Mechanism)** *The optimal transport plan of the elastic optimal transport problem in Equation 13 is equivalent to that of its corresponding constrained worst transport problem in Equation 15, i.e.,*

$$\gamma^* = \arg\min_{\gamma \in \Gamma_e(\mu, \nu)} \sum_{i,j} \mathcal{C}_{ij} \gamma_{ij} = \arg\max_{\gamma \in \Gamma_{\mathcal{C}}(\mu, \nu)} \sum_{i,j} -\mathcal{C}_{ij}^- \gamma_{ij}. \tag{17}$$

Theorem 2 (proof in Appendix A.1.2) gives insights into the mass transportation mechanism of the elastic optimal transport problem. The source and target areas can be divided into active and inactive regions according the sign of ground cost. There is no mass transportation between the source and the target areas when their cost are positive. The mass allocation between the areas with negative cost acts like that of the constrained worst transport problem, where the optimal transport plan satisfies either left or right marginal equality constraint rather than both inequality constraints.

From a pragmatic point of view, Theorem 2 indicates that elastic optimal transport provides a robust solution to handle noise, outliers, and distribution shifts among data. Suppose that the items of $i^{th}$ row (or $j^{th}$ column) in the cost matrix are all positive, then there is no mass transported from $x_i$ (or to $z_j$) according to Theorem 2, i.e., $\sum_j \gamma_{ij}^* = 0 < \mu_i$ (or $\sum_i \gamma_{ij}^* = 0 < \nu_j$). It suggests that $x_i$ (or $z_j$) is likely to be outliers or noisy point. The strength of elastic optimal transport is that it is capable of filtering out outliers automatically by enabling adaptive mass allocation. According to the above discussion, it is straightforward to attain the following corollary (the proof is omitted):

**Corollary 1 (Optimal Transport Mass)** *For the elastic optimal transport problem, the total mass $s$ of the optimal transport plan satisfies:*

$$s = \sum_{ij} \gamma_{ij}^* \leq \min \left\{ \sum_{i \in I} \mu_i, \sum_{j \in J} \nu_j \right\} \leq \min \left\{ \|\mu\|_1, \|\nu\|_1 \right\} \tag{18}$$

*where $I$ (or $J$) is the set of row (or column) in the cost matrix who has at least one negative item.*

## 3 APPLICATION TO DOMAIN ADAPTATION

Optimal transport plays fundamental roles in many machine learning applications. As a crucial task of machine learning, domain adaptation aims to estimate a transferable model for target domain by exploiting source domain data in the presence of domain shift. We focus on both unsupervised and partial domain adaptation, where no label is available in the target domain.

Let $\mathcal{X} = \{x_i\}_{i=1}^n$ be a set of data samples drawn from a distribution $\mu$ on the source domain, associated with a set of class labels $\mathcal{Y} = \{y_i\}_{i=1}^n$, and $\mathcal{Z} = \{z_j\}_{j=1}^m$ a set of data samples drawn from a distribution $\nu$ on the target domain. The empirical distribution $\mu$ for source domain is formulated as $\mu = \sum_{i=1}^n p_i \delta_{x_i}$ where $\delta_{x_i}$ is the Dirac function located at $x_i$, and $p_i$ is probability mass associated with $x_i$ and belongs to the probability simplex, i.e. $\sum_{i=1}^n p_i = 1$. The distribution $\nu$ for target domain can be formulated similarly.

We propose the generalized unsupervised domain adaptation model based on elastic optimal transport, called ELOT. Let $g$ be the feature extractor that maps the data samples into latent feature space, and $f$ the classifier which maps the latent feature space into the label space. Both the ground cost matrix $\mathcal{C}(g, f)$ and the empirical classification loss $\mathcal{L}(g, f)$ depend on the extractor $g$ and the classifier $f$. The objective of ELOT is to minimize the elastic optimal transport distance between the source domain and target domain, as well as the empirical classification loss on the source domain, i.e.,

$$\min_{\gamma \in \Pi_e(\mu, \nu), f, g} < \gamma, \mathcal{C}(g, f) > + \mathcal{L}(g, f) \tag{19}$$

where $\mathcal{C}(g, f) \in \mathbb{R}_{\pm}^{n \times m}$ and

$$\Pi_e(\mu, \nu) = \{\gamma \in \mathbb{R}_+^{n \times m} \mid \gamma \mathbf{1}_m \leq \mu, \gamma^T \mathbf{1}_n \leq \nu\}. \tag{20}$$

The strength of using elastic optimal transport for domain adaptation is its capability of transporting partial mass adaptively from source domain to target domain. Due to the domain shift, full-mass conservation may lead to fitting dissimilar pairs (and noise or outliers) between source and target domains. Also, since it is unknown to what extent the two domains are related, fixed-mass conservation is too restrictive. Therefore, the most suitable way is to fit similar sample pairs between domains while neglecting dissimilar pairs and outliers via adaptive transportation of partial masses.

Specifically, we provide an instantiation of ELOT as follows. Nevertheless, there are a variety of choices of ground costs, depending on the problems themselves.

$$\min_{\gamma, g, f} \sum_{i,j} \gamma_{ij} \left[ \alpha \big\| g(x_i) - g(z_j) \big\|^2 - \beta y_i \tanh \Big( f\big(g(z_j)\big) \Big) \right] - \sum_i y_i \log \Big( f\big(g(x_i)\big) \Big) \tag{21}$$

where $\gamma \in \Pi_e(\mu, \nu)$, and $tanh$ is the hyperbolic Tangent function. $\alpha$ and $\beta$ are non-negative coefficients to balance the impact of feature-wise cost and label-wise cost. Here we use cross-entropy loss as the empirical classification loss.

ELOT requires the costs to be mixed-sign to avoid the potential degeneracy. This could be a limitation in some tasks, but ELOT is widely applicable to many scenarios such as domain adaptation where the cost could be induced in multiple feature spaces (e.g., original feature space and latent feature space, or marginal distribution and conditional distribution). Specifically, the underlying idea in constructing the ground cost matrix $\mathcal{C}$ is to align the domains in the feature space and label space simultaneously. Since the target labels are unknown, we use the surrogate version $f(g(z))$ which is a common practice to use the pseudo labels. The ground cost consists of two types of cost, i.e. feature-wise cost and label-wise cost. The intuition is that the more similar the pair is in both feature and label space, the more mass transported between them.

Note that ELOT is general and many kinds of ground cost functions could be used in ELOT. For example, one may define the alternative ground cost function as follows:

$$\mathcal{C}_{ij} = \alpha \big\| g(x_i) - g(z_j) \big\|^2 - \beta \langle f\big(g(x_i)\big), f\big(g(z_j)\big) \rangle. \tag{22}$$

It is worth mentioning we use exactly the same number of hyper-parameters as JUMBOT (Fatras et al., 2021) and m-POT (Nguyen et al., 2022) to construct the cost function. Since one may move the coefficient $\alpha$ from the cost term to the cross-entropy loss term as usually done to achieve the same effect, there is actually only one hyper-parameter $\beta$ used in the cost function. Besides, both JUMBOT and m-POT need extra trade-off parameters $(s, \tau_1, \tau_2)$ to set either the marginal penalization coefficient or a priori amount of mass to be transferred, which remains a challenging issue. Therefore, the advantage of ELOT over partial OT or unbalanced OT is that it not only achieves adaptive-mass transport but also requires less trade-off coefficients.

It is unavoidable to tune the hyper-parameter $\beta$ in these cases since this hyper-parameter is a part of the cost structure, depending on the problem itself. As usually, it could be tuned via cross-validation. Hence, ELOT relies on the cost structure for adaptive-mass transport, rather than playing the exchange game of hyper-parameter.

The detailed ELOT algorithm is presented in Appendix A.2.

## 4 EXPERIMENTS

We evaluate our method on both unsupervised domain adaptation and partial domain adaptation tasks in comparison with the state-of-the-art algorithms. The implementation details are introduced in Appendix A.4.1.

### 4.1 DATASETS

**VisDA** (Peng et al., 2017) is a large-scale dataset for unsupervised domain adaptation. It contains 152,397 synthetic images as the source domain and 55,388 real-world images as the target domain. The two domains share 12 object categories. Following the common setting (Fatras et al., 2021; Nguyen et al., 2022), we evaluate all methods on VisDA validation set. **Office-Home** (Venkateswara et al., 2017) contains 15,500 images from four domains: Artistic images (A), ClipArt (C), Product images (P) and Real-World (R). For each domain, it consists of 65 object categories that are common in home and office scenarios. All methods are evaluated on 12 adaptation tasks. **Office-31** (Saenko et al., 2010) consists of 4652 images from 31 categories, collected from three domains including Amazon (2817 images), Webcam (795 images) and DSLR (498 images), respectively. There are totally 6 adaptation tasks for evaluation. **DomainNet** (Peng et al., 2019) is a large-scale dataset with six distinct domains and approximately 0.6 million images distributed among 345 categories. We follow Gu et al. (2024) to adopt four domains including Clipart (C), Painting (P), Real (R), and Sketch (S)) with 126 classes for partial domain adaptation task. All above datasets are widely used for the evaluation of domain adaptation algorithms.

### 4.2 UNSUPERVISED DOMAIN ADAPTATION

We compare ELOT with a variety of unsupervised domain adaptation algorithms: 1) OT-based methods such as ROT (Balaji et al., 2020), DeepJDOT (Damodaran et al., 2018), JUMBOT (Fatras et al., 2021), and m-POT (Nguyen et al., 2022); and 2) Non-OT-based methods such as DANN (Ganin et al., 2016), CDAN (Long et al., 2017a), ALDA (Chen et al., 2020), and CaCo (Huang et al., 2022).

DEEPJDOT (Damodaran et al., 2018) is the deep learning-based extension of JDOT (Courty et al., 2017a) which is a joint distribution optimal transport method. ROT (Balaji et al., 2020) followed the unbalanced optimal transport formulation while keeping the $f$-divergence relaxations of marginal distributions as inequality constraints. JUMBOT (Fatras et al., 2021) adopted the unbalanced optimal transport to alleviate the issue of undesired matching during the mini-batch sampling. In contrast, m-POT (Nguyen et al., 2022) used partial optimal transport to mitigate the misspecified mappings by limiting the amount of masses. Domain Adversarial Neural Network (DANN) (Ganin et al., 2016) adversarially learned a feature extractor and a domain discriminator. Conditional Domain Adversarial Network (CDAN) (Long et al., 2017a) applied a conditional domain discriminator instead. ALDA (Chen et al., 2020) combined self-training and adversarial training for noise-correction domain discrimination. The contrastive-learning based method CaCo (Huang et al., 2022) adopted the category contrastive loss for adaptation. The more discussion of related work on domain adaptation and distinctive characteristic of ELOT are given in Appendix A.5.

The main goal of our experiments is to verify the superiority of elastic optimal transport over the classical OT theory such as partial OT and unbalanced OT. We are not aiming to pursue SOTA performance on these datasets, which could be attained by using more advanced backbones. For fair comparison, the backbones of all the methods are based on the deep neural network ResNet-50 (He et al., 2016) pretrained on ImageNet. We conduct each experiment three times and report the average Accuracy score (in %) and standard deviation. The accuracies of the comparison methods are reproduced, or quoted from JUMBOT (Fatras et al., 2021), m-POT (Nguyen et al., 2022) or their papers unless otherwise stated. The standard deviations for comparison methods are shown whenever available in their papers.

Tables 1, 2, 3 show the proposed ELOT method significantly outperforms the comparison baselines. The bold and underlined accuracies represent the best and the second best performance. On the large-scale VisDA dataset which is one or two order of magnitude higher than the other two datasets, ELOT beats the runner-up method m-POT by a large margin of 2.73. On Office-31 dataset, ELOT performs consistently better than the comparison methods on all 6 adaptation tasks. For Office-Home dataset, ELOT also achieve higher accuracies in 12 out of 12 adaptation scenarios.

We take a closer look at the unsupervised domain adaptation methods based on optimal transport. DeepJDOT (Damodaran et al., 2018) adopted Kantorovich's optimal transport to align the joint features/labels distribution. JUMBOT (Fatras et al., 2021) and m-POT (Nguyen et al., 2022) followed DeepJDOT to align joint distributions, while using unbalanced optimal transport and partial optimal transport respectively instead of Kantorovich's optimal transport. JUMBOT and m-POT leveraged the geometrically robust versions of OT to alleviate the influence of undesired coupling between samples, leading to better performance than DeepJDOT. ELOT consistently outperforms the above OT based methods on the large-scale and medium-scale datasets including VisDA, Office-31, and Office-Home. It demonstrates the superiority of elastic optimal transport in domain alignment by throwing away the full-mass or fixed-mass constraints. The strengthen of ELOT is that it is adept at making choices on whether the pairs should be mapped or not, and how much masses should be transported, by taking the domain shift into full consideration. Moreover, ELOT relies on the geometry structure of the problem itself to transport suitable masses adaptively, providing a better choice than the fixed-mass strategy for domain adaptation.

Table 1: Classification accuracy on VisDA (ResNet-50) for *unsupervised* domain adaptation tasks, where (*) and (#) denote the results quoted from m-POT and JUMBOT.

| Method | Accuracy |
|---|---|
| DANN[*] | 67.63±0.34 |
| CDAN[#] | 70.10 |
| ALDA[*] | 71.22±0.12 |
| ROT[#] | 66.30 |
| DeepJDOT[#] | 68.00 |
| JUMBOT[#] | 72.50 |
| m-POT[*] | 73.59±0.15 |
| **ELOT (ours)** | **76.32**±0.28 |

Table 2: Classification accuracy on Office-31 (ResNet-50) for *unsupervised* domain adaptation tasks. The results reproduced from m-POT are denoted with (○).

| Method | A→W | D→W | W→D | A→D | D→A | W→A | Avg |
|---|---|---|---|---|---|---|---|
| ResNet-50 | 68.4 | 96.7 | 99.3 | 68.9 | 62.5 | 60.7 | 76.1 |
| DANN | 82.0 | 96.9 | 99.1 | 79.7 | 68.2 | 67.4 | 82.2 |
| CDAN | 93.1 | 98.6 | **100.0** | 92.9 | 71.0 | 69.3 | 87.5 |
| ALDA | 95.6 | 97.7 | **100.0** | **94.0** | 72.2 | 72.5 | 88.7 |
| CaCo | 89.7 | 98.4 | **100.0** | 91.7 | 73.1 | 72.8 | 87.6 |
| DeepJDOT (○) | 87.8±0.2 | 97.9±0.3 | 99.8±0.1 | 88.7±0.1 | 70.8±0.3 | 71.3±0.2 | 86.1 |
| JUMBOT (○) | 91.5±0.4 | 98.5±0.2 | **100.0**±0 | 89.4±0.3 | 70.2±0.2 | 86.4 |
| m-POT (○) | 93.7±0.3 | **99.1**±0.1 | **100.0**±0 | 93.3±0.2 | 70.9±0.4 | 72.5±0.1 | 88.3 |
| **ELOT(ours)** | **96.0**±0.3 | **99.1**±0.1 | **100.0**±0 | **94.0**±0.2 | **77.4**±0.1 | **76.4**±0.1 | **90.5** |

Table 3: Classification accuracy on Office-Home (ResNet-50) for *unsupervised* domain adaptation tasks, where (*) denotes the results quoted from m-POT.

| Task | A-C | A-P | A-R | C-A | C-P | C-R | P-A | P-C | P-R | R-A | R-C | R-P | Avg |
|---|---|---|---|---|---|---|---|---|---|---|---|---|---|
| ResNet-50(*) | 34.90 | 50.00 | 58.00 | 37.40 | 41.90 | 46.20 | 38.50 | 31.20 | 60.40 | 53.90 | 41.20 | 59.90 | 46.10 |
| DANN(*) | 47.92 | 67.08 | 74.85 | 53.80 | 63.47 | 66.42 | 52.99 | 44.35 | 74.43 | 65.53 | 52.96 | 79.41 | 61.93 |
| CDAN(*) | 52.50 | 71.40 | 76.10 | 59.70 | 69.90 | 71.50 | 58.70 | 50.30 | 77.50 | 70.50 | 57.90 | 83.50 | 66.60 |
| ALDA(*) | 54.04 | 74.89 | 77.14 | 61.37 | 70.62 | 72.75 | 60.32 | 51.03 | 76.66 | 67.90 | 55.94 | 81.87 | 67.04 |
| ROT(*) | 47.20 | 70.80 | 76.40 | 58.60 | 68.10 | 70.20 | 56.50 | 45.00 | 75.80 | 69.40 | 52.10 | 80.60 | 64.30 |
| DeepJDOT(*) | 51.75 | 70.01 | 75.59 | 59.60 | 66.46 | 70.07 | 57.60 | 47.88 | 75.29 | 66.82 | 55.71 | 78.11 | 64.59 |
| JUMBOT(*) | 54.99 | 74.45 | 80.78 | 65.66 | 74.93 | 74.91 | 64.70 | 53.42 | 80.01 | 74.58 | 59.88 | 83.73 | 70.17 |
| m-POT(*) | 55.65 | 73.80 | 80.76 | 66.34 | 74.88 | 76.16 | 64.46 | 53.38 | 80.60 | 74.55 | 59.71 | 83.81 | 70.34 |
| **ELOT(ours)** | **56.79**±0.2 | **77.41**±0.1 | **82.18**±0.2 | **70.26**±0.2 | **75.03**±0.3 | **78.03**±0.2 | **66.64**±0.1 | **53.68**±0.2 | **81.17**±0.0 | **75.39**±0.2 | **60.26**±0.2 | **84.16**±0.1 | **71.75** |

## 4.3 PARTIAL DOMAIN ADAPTATION

In order to further demonstrate its generality and robustness, we evaluate ELOT on partial domain adaptation tasks, where the target labels are the subset of the source labels. Samples belonging to the missing classes become outliers which poses a great challenge to the robustness of the methods.

Table 4: Classification accuracy on Office-Home (ResNet-50) for *partial* domain adaptation tasks.

| Task | A-C | A-P | A-R | C-A | C-P | C-R | P-A | P-C | P-R | R-A | R-C | R-P | Avg |
|---|---|---|---|---|---|---|---|---|---|---|---|---|---|
| ResNet-50($*$) | 46.30 | 67.50 | 75.90 | 59.10 | 59.90 | 62.70 | 58.20 | 41.80 | 74.90 | 67.40 | 48.20 | 74.20 | 61.40 |
| PADA($*$) | 51.90 | 67.00 | 78.70 | 52.20 | 53.80 | 59.00 | 52.60 | 43.20 | 78.80 | 73.70 | 56.60 | 77.10 | 62.10 |
| ETN($*$) | 59.20 | 77.00 | 79.50 | 62.90 | 65.70 | 75.00 | 68.30 | 55.40 | 84.40 | 75.70 | 57.70 | 84.50 | 70.40 |
| BA3US($*$) | 59.34 | 78.73 | 88.42 | 72.70 | 72.34 | 83.54 | 73.19 | 60.20 | 85.92 | 79.13 | 63.00 | 85.90 | 75.20 |
| DeepJDOT($*$) | 48.00 | 65.99 | 77.47 | 59.23 | 57.85 | 66.57 | 58.43 | 45.25 | 74.10 | 68.08 | 49.89 | 74.25 | 62.09 |
| JUMBOT($*$) | 61.53 | 80.34 | 85.33 | 75.60 | 72.89 | 79.79 | 74.56 | 61.95 | 86.49 | 80.78 | 67.38 | 84.89 | 75.96 |
| m-POT($*$) | **64.60** | 80.62 | 87.17 | 76.43 | 77.61 | **83.58** | 77.07 | 63.74 | 87.63 | 81.42 | 68.50 | **87.38** | 77.98 |
| **ELOT(ours)** | 64.54$\pm$0.1 | **86.16**$\pm$0.1 | **88.46**$\pm$0.2 | **77.87**$\pm$0.1 | **78.43**$\pm$0.1 | 83.55$\pm$0.2 | **80.44**$\pm$0.2 | **65.19**$\pm$0.1 | **87.80**$\pm$0.1 | **82.19**$\pm$0.1 | **69.08**$\pm$0.2 | 86.59$\pm$0.1 | **79.19** |

Table 5: Classification accuracy on DomainNet (ResNet-50) for *partial* domain adaptation tasks.

| Method | C$\rightarrow$P | C$\rightarrow$R | C$\rightarrow$S | S$\rightarrow$C | S$\rightarrow$R | S$\rightarrow$P | Avg |
|---|---|---|---|---|---|---|---|
| ResNet-50 | 41.2 | 60.0 | 42.1 | 45.4 | 39.3 | 49.8 | 46.3 |
| DANN | 27.8 | 36.6 | 29.9 | 25.8 | 29.5 | 32.7 | 30.4 |
| CDAN | 37.5 | 48.3 | 46.6 | 35.4 | 38.5 | 43.6 | 41.7 |
| PADA | 22.5 | 32.9 | 30.0 | 17.5 | 23.9 | 26.9 | 25.6 |
| BA3US | 42.9 | 54.7 | 53.8 | 50.4 | 42.7 | 49.7 | 49.0 |
| STCPDA | 65.1 | 69.6 | **69.6** | 64.4 | 60.7 | 67.8 | 66.2 |
| ARPM | 67.9 | **79.8** | 66.3 | 62.5 | 64.8 | 71.7 | 68.8 |
| **ELOT (Ours)** | **68.8** | 77.7 | 68.3 | **68.0** | **69.9** | **75.9** | **71.4** |

In order to verify that various cost functions could be used in ELOT, we use the alternative ground cost function (22) in partial domain adaptation task. Our empirical study shows both the cost functions in Equations 21 and 22 attain the similar performance. Therefore, we report the results using the cost function 22 only.

We evaluate ELOT on Office-Home dataset, following BA3US (Liang et al., 2020) to select the first 25 out of 65 categories (in alphabetic order) in each domain as a partial target domain. ELOT is compared with a variety of non-OT-based methods including PADA (Cao et al., 2018), ETN (Cao et al., 2019), BA3US (Liang et al., 2020), and OT-based methods including DeepJDOT (Damodaran et al., 2018), JUMBOT (Fatras et al., 2021), and m-POT (Nguyen et al., 2022). The results are shown in Table 4.

For a fair comparison, we do not use ten crop technique on testsets, following from JUMBOT (Fatras et al., 2021) and m-POT (Nguyen et al., 2022). The network architecture and training details are similar to the unsupervised domain adaptation setting on the Office-Home dataset. The experiments are trained for 5000 iterations. The batch size is 65 which is consistent with the OT-based methods. The trade-off coefficients are set as follows: $\alpha = 0.01, \beta = 4.5, \epsilon = 1$. The results of the comparison methods are quoted from m-POT (Nguyen et al., 2022) and JUMBOT (Fatras et al., 2021).

Table 4 reinforces the superiority of ELOT over the comparison methods. ELOT achieves the highest accuracy on 9 out of 12 partial domain adaptation tasks, and obtains the comparable results with the strongest competitors on the remaining tasks. Partial domain adaptation provides an ideal test bed for evaluating the robustness of the methods because the outliers belonging to the missing labels could cause negative transfer. The existence of a considerable number of outliers would amplify the risk of mapping undesired pairs such the samples with different classes. The experiment results suggest that ELOT provides a more robust solution for domain alignment than the conventional OT methods. Furthermore, ELOT not only yields adaptive-mass transport but also uses less hyper-parameter. In contrast, both unbalanced OT (e.g., JUMBOT) and partial OT (e.g., m-POT) need extra trade-off parameters to set either the marginal penalization coefficient or a priori amount of mass to be transferred, which remains a nontrivial issue.

Table 5 reports the average classification accuracy for six partial domain adaptation tasks on the DomainNet dataset. DomainNet is a large-scale challenging dataset with 345 classes. We follow ARPM (Gu et al., 2024) and use the first 40 classes in alphabetical order to build the target domain in each task. ELOT significantly outperforms the comparison methods such as the recent methods including STCPDA (He et al., 2024) and ARPM (Gu et al., 2024), demonstrating its superiority on the large-scale challenging dataset.

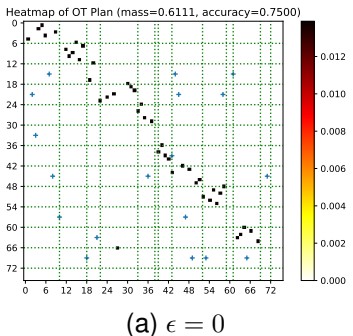 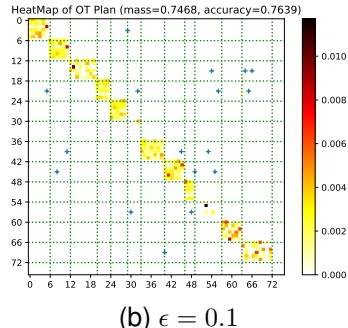

(a) $\epsilon = 0$            (b) $\epsilon = 0.1$

Figure 1: Heatmap of optimal transport plan. The masses are allocated along diagonal blocks, aligning labels between domains.

### 4.4 HEATMAP OF TRANSPORT PLAN

We go deep into the transport plan to investigate how the masses are adaptively transported from source domain to target domain. For VisDA dataset, Figure 1 plots the heatmaps of transport plans in a mini-batch with batch size $b = 72$. Similar to the previous work (Fatras et al., 2021; Nguyen et al., 2022), we adopt the stratified sampling to select a mini-batch of source samples so that each class has the same number of samples. For the target domain, a mini-batch samples are randomly selected since the target labels are unavailable in training.

Each row (or column) corresponds to one source (or target) sample. The source (or target) samples are reordered into clusters by the order of labels (or pseudo labels). Therefore, the $72 \times 72$ transport plan matrix is partitioned into $12 \times 12$ blocks since there are 12 classes, except for the cases some target labels are absent due to random sampling. As expected, the masses are almost allocated along the diagonal blocks, aligning labels between source and target domains. It suggests that the optimal transport in ELOT is class-aware. Also, since pseudo labels are used, it is likely that some predictions are wrong. To check which prediction is wrong, we use '+' located in the row cluster to indicate the true label (corresponding to the row cluster) of the target sample.

Figure 1 plots the heatmaps with $\epsilon = 0$ and $\epsilon = 0.1$ respectively. It intuitively shows that the transport plan becomes sparser as the entropy-regularized coefficient $\epsilon$ increases. As shown on the top of the figures, the total masses transported from source domain to target domain are 0.6111 and 0.7468 for $\epsilon = 0$ and $\epsilon = 0.1$, respectively. We compare the optimal transport masses obtained by our method with those of m-POT (Nguyen et al., 2022). An interesting finding is that m-POT achieved the best accuracy when the fraction of masses is set to 0.75 (see Figure 2 of (Nguyen et al., 2022)), which is very close to 0.7468 obtained by our method. However it is challenging to pre-define the optimal fraction of masses in m-POT. In contrast, our method can automatically learn the optimal fraction of masses to be transported, leading to an elegant solution for adaptive matching.

In summary, the heatmap intuitively verified that ELOT transports masses between domains in an adaptive and class-aware way.

The more experiment results such as hyper-parameter sensitivity analysis and T-SNE visualization are discussed in Appendices A.4.2 and A.4.3.

## 5 CONCLUSION

We propose a novel formulation of optimal transport along with the solid theoretical analysis to remedy the limitation of classical optimal transport. Elastic optimal transport leverages the geometry structure of the problem itself to transport adaptive-mass rather than full or fixed masses. It provides a flexible and powerful tool to model the problem of aligning the probability distributions adaptively by respecting the ubiquity of noises, outliers, and divergences existing in the data. Far beyond domain adaptation, we believe that elastic optimal transport opens the pathway to unlock the problems of adaptive distribution matching in a variety of areas such as artificial intelligence, data science, bioinformatics, economics, operations research, etc.

## ACKNOWLEDGMENTS

This work is partially supported by National Natural Science Foundation of China (No 62211530114) and Basic and Applied Basic Research Foundation of Guangdong (No 2023A1515010750).

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

# A  APPENDIX

## A.1  THE PROOFS OF THEOREMS

### A.1.1  PROOF OF THEOREM 1

**Proof 1** *We get rid of the inequality constraints of the elastic optimal transport problem defined in Equation 6 by adding the slack variables:*

$$\gamma \mathbf{1}_m + b = \mu \tag{23}$$

$$\gamma^{\mathrm{T}} \mathbf{1}_n + a = \nu \tag{24}$$

*where $a \in \mathbb{R}_+^{m \times 1}$ and $b \in \mathbb{R}_+^{n \times 1}$ are slack variables. Denote the transported mass by $s = \sum_{i=1}^{n} \sum_{j=1}^{m} \gamma_{ij}$. Note that $s$ is adaptively learned instead of user-defined or fixed. Based on the above two equations, we have*

$$s + \|b\|_1 = \|\mu\|_1,$$
$$s + \|a\|_1 = \|\nu\|_1.$$

*Denote the extended transport plan by*

$$\bar{\gamma} = \begin{bmatrix} \gamma & b \\ a^{\mathrm{T}} & s \end{bmatrix}. \tag{25}$$

*Next, we show that both the elastic optimal transport problem and its reformulation have the equivalent optimal transport plan.*

$$
\begin{aligned}
\bar{\boldsymbol{\gamma}}^* &= \underset{\bar{\gamma}\in\Pi(\bar{\mu},\bar{\nu})}{\arg\min} \sum_{i=1}^{n+1}\sum_{j=1}^{m+1} \bar{\mathcal{C}}_{ij}\bar{\gamma}_{ij} \\
&= \underset{\bar{\gamma}\in\Pi(\bar{\mu},\bar{\nu})}{\arg\min} \sum_{i=1}^{n}\sum_{j=1}^{m} \mathcal{C}_{ij}\bar{\gamma}_{ij} + \sigma\Big(\|a\|_1 + \|b\|_1 + 2s\Big) \\
&= \underset{\bar{\gamma}\in\Pi(\bar{\mu},\bar{\nu})}{\arg\min} \sum_{i=1}^{n}\sum_{j=1}^{m} \mathcal{C}_{ij}\bar{\gamma}_{ij} + \sigma\Big(\|\mu\|_1 + \|\nu\|_1\Big) \\
&= \underset{\bar{\gamma}\in\Pi(\bar{\mu},\bar{\nu})}{\arg\min} \sum_{i=1}^{n}\sum_{j=1}^{m} \mathcal{C}_{ij}\bar{\gamma}_{ij} + Const \\
&= \underset{\bar{\gamma}\in\Pi(\bar{\mu},\bar{\nu})}{\arg\min} \sum_{i=1}^{n}\sum_{j=1}^{m} \mathcal{C}_{ij}\bar{\gamma}_{ij}.
\end{aligned}
$$

*It suggests that the optimal transport plan $\bar{\boldsymbol{\gamma}}^*$ for the reformulated problem is controlled by the cost matrix $\mathcal{C}$ rather than the augmented one $\bar{\mathcal{C}}$. Assume $\bar{\boldsymbol{\gamma}}_o^*$ is the matrix $\bar{\boldsymbol{\gamma}}^*$ with the last row and column removed. Since $\bar{\boldsymbol{\gamma}}^* \in \Pi(\bar{\mu},\bar{\nu})$, we have $\bar{\boldsymbol{\gamma}}_o^* \in \Pi_e(\mu,\nu)$ according to Equations 23, 24, and 25. Following the last equation of the above derivations, we prove the equivalence of the optimal transport plans:*

$$
\bar{\boldsymbol{\gamma}}_o^* = \underset{\gamma\in\Pi_e(\mu,\nu)}{\arg\min} \sum_{i=1}^{n}\sum_{j=1}^{m} \mathcal{C}_{ij}\gamma_{ij} = \boldsymbol{\gamma}^*.
$$

*Meanwhile, according to the above derivations, we have the result that the difference between the optimal transport distances is a constant:*

$$
\mathbf{W}(\bar{\mu},\bar{\nu}) - \mathbf{W}(\mu,\nu) = \sigma\Big(\|\boldsymbol{\mu}\|_1 + \|\boldsymbol{\nu}\|_1\Big) = Const
$$

*which can be ignored in optimization.*

### A.1.2  PROOF OF THEOREM 2

**Proof 2** *Let $\gamma^*$ be the optimizer for the elastic optimal transport problem. We first derive two propositions:*
ⓐ *For the positive cost $\mathcal{C}_{ij} > 0$, we have $\gamma_{ij}^* = 0$.*
ⓑ *For the negative cost $\mathcal{C}_{ij} < 0$, we have*

$$
\sum_{j}\gamma_{ij}^* = \mu_i \ \textbf{ or } \ \sum_{i}\gamma_{ij}^* = \nu_j. \tag{26}
$$

*We prove the proposition ⓑ by reduction to absurdity, and the proposition ⓐ can be proved in a similar way. Assume that both equations in Equation 26 do not hold for $\mathcal{C}_{ij} < 0$. Let's set*

$$
\triangle m = min\Big\{\mu_i - \sum_{j}\gamma_{ij}^*, \ \nu_j - \sum_{i}\gamma_{ij}^*\Big\}.
$$

*One can increase the total transport mass by $\triangle m$ such that at least one of the equations in Equation 26 holds. Then the objective of the elastic optimal transport problem will decrease by $-\mathcal{C}_{ij} \cdot \triangle m$. Note that $-\mathcal{C}_{ij} \cdot \triangle m > 0$, therefore we obtain a better optimal transport plan, which contradicts with the premise that $\gamma^*$ is the optimizer.*
*Then we divide the elastic optimal transport problem in Equation 13 into two sub-problems according to the sign of ground cost, and show the equivalence between the elastic optimal transport*

---

**Algorithm 1** The **ELOT** Algorithm

---

**Input:** Source data $\mathcal{X} = \{x_i\}_{i=1}^n$ with label set $\mathcal{Y} = \{y_i\}_{i=1}^n$, target data $\mathcal{Z} = \{z_j\}_{j=1}^m$, trade-off parameters $\alpha$ and $\beta$, batch-size $b$, and maximum epoch $T$.
**Output:** The learned extractor $g$ and classifier $f$.
 1: Randomly initialize the classifier $f$.
 2: **for** $t = 1$ to $T$ **do**
 3:     Randomly sample the mini-batch samples from source data $\{(\mathbf{x}_i, \mathbf{y}_i)\}_{i=1}^n$ and target data $\{\mathbf{z}_j\}_{j=1}^m$, respectively.
 4:     Fix extractor $g$ and classifier $f$.
 5:     Compute the ground cost matrix $\mathcal{C}$.
 6:     Obtain the optimal transport plan $\gamma$ by solving the ELOT problem defined in Equation 21.
 7:     Fix transport plan $\gamma$.
 8:     Compute the total loss.
 9:     Update extractor $g$ and classifier $f$ via SGD algorithm.
10: **end for**

---

*problem and its corresponding constrained worst transport problem as follows:*

$$
\begin{aligned}
\gamma^* &= \underset{\gamma \in \Gamma_e(\mu,\nu)}{\arg\min} \sum_{i,j} \mathcal{C}_{ij}\gamma_{ij} \\
&= \underset{\gamma \in \Gamma_e(\mu,\nu)}{\arg\min} \sum_{i,j} \mathcal{C}_{ij}^+\gamma_{ij} + \sum_{i,j} \mathcal{C}_{ij}^-\gamma_{ij} \\
&\overset{\text{ⓐ}}{=} \underset{\gamma \in \Gamma_e(\mu,\nu)}{\arg\min} \sum_{i,j} \mathcal{C}_{ij}^-\gamma_{ij} \\
&\overset{\text{ⓑ}}{=} \underset{\gamma \in \Gamma_\mathcal{C}(\mu,\nu)}{\arg\min} \sum_{i,j} \mathcal{C}_{ij}^-\gamma_{ij} \\
&= \underset{\gamma \in \Gamma_\mathcal{C}(\mu,\nu)}{\arg\max} \sum_{i,j} -\mathcal{C}_{ij}^-\gamma_{ij}
\end{aligned}
\tag{27}
$$

*where the third and fourth equations follow from the propositions ⓐ and ⓑ, respectively.*

## A.2    THE ELOT ALGORITHM

Algorithm 1 adopts a mini-batch sampling strategy to solve the elastic optimal transport problem, enabling it scalable to large datasets. In each mini-batch, it iteratively updates the transport plan $\gamma$, and the feature extractor $g$ and the classifier $f$. First, when the extractor $g$ and the classifier $f$ are fixed, the problem defined in Equation 21 reduces to the standard elastic optimal transport. According to Theorem 1, the optimal transport plan can be obtained by solving the reformulation of the elastic optimal transport problem. We adopt the linear programming or the Sinkhorn-Knopp algorithm (Cuturi, 2013) to solve the reformulated problem. Second, fixing the transport plan $\gamma$, the extractor $g$ and the classifier $f$ can be updated with the stochastic gradient descent algorithm (SGD).

## A.3    RELATIONS WITH EXISTING OPTIMAL TRANSPORT

We clarify that ELOT is fundamentally different from unbalanced optimal transport (UOT) and partial optimal transport (POT), highlighting the novelty of ELOT.

### A.3.1    ELOT VS UOT

We would like to clarify that ELOT is not a limiting case of unbalanced optimal transport. As the penalty coefficients approach zero, i.e., $\tau_1 \to 0$ and $\tau_2 \to 0$, allowing relaxed inequalities, UOT will lead to the degenerate solution with zero transport plan due to the nonnegative constraint of costs. Even though UOT is allowed to accept the mixed-sign cost (which means the nonnegative constraint on costs is violated), it will lead to the solution with infinite mass flowing on negative-cost entries due to the overly relaxed inequalities $\tau_1 \mathcal{D}_\phi(\gamma 1_m \| \mu) + \tau_2 \mathcal{D}_\phi(\gamma^T 1_n \| \nu)$ in the case of $\tau_1 \to 0$ and $\tau_2 \to 0$. In contrast, ELOT depends on the combination of the marginal inequality constraints and mixed-sign cost to achieve adaptive-mass transport.

Therefore, ELOT is certainly not the limiting case of unbalanced OT. Instead, ELOT is fundamentally distinctive from unbalanced optimal transport. First, the problem formulations are different. Second, the mass transportation mechanism is different, as indicated the theoretical analysis. ELOT can achieve adaptive-mass transport, relying on the combination of the marginal inequality constraints and mixed-sign cost, while UOT depends on users to specify the hyper-parameters of soft constraints.

### A.3.2 ELOT VS POT

We go deep into the relation and difference between ELOT and partial optimal transport, and highlight the distinctive advantages of ELOT against POT.

The continuous formulations of optimal transport are used in this subsection. Suppose $X, Z \subset \mathbb{R}^d$ are domains in Euclidean space. Let $c(x, z)$ be the bounded continuous cost function which tells how much it costs to move one unit of mass from location $x \in \mathcal{X}$ to location $z \in \mathcal{Z}$. The transport plan $\gamma(x, z)$ measures the amount of mass transferred from location $x$ to location $z$. Without loss generalization, let's assume that both $\mu$ and $\nu$ are probability measures such that $\int_X \mu(x)dx = \int_Z \nu(z)dz = 1$ here for simplicity.

Partial optimal transport (Caffarelli & McCann, 2010) is then formulated as

$$\min_{\substack{\gamma \in \Gamma_{\leq}(\mu,\nu), \\ \int_{X \times Z} d\gamma(x,z)=m}} \int_{X \times Z} c^+(x, z)d\gamma(x, z) \tag{28}$$

where the cost function $c^+(x, z)$ is non-negative, and $m$ is the user-specified mass budget. The set of admissible transport plans is denoted by

$$\Gamma_{\leq}(\mu, \nu) = \left\{ \gamma \geq 0 \middle| \int_Z \gamma(x, z)dz \leq \mu(x), \int_X \gamma(x, z)dx \leq \nu(z) \right\} \tag{29}$$

Caffarelli and McCann introduced a Lagrange multiplier $\lambda_m \geq 0$ conjugate to the fixed-mass constraint $\int_{X \times Z} d\gamma(x, z) = m$ and reformulated the POT problem as

$$\min_{\gamma \in \Gamma_{\leq}(\mu,\nu)} \int_{X \times Z} \left[ c^+(x, z) - \lambda_m \right] d\gamma(x, z). \tag{30}$$

Likewise, elastic optimal transport defined in continuous formulation can be reformulated as

$$\min_{\gamma \in \Gamma_{\leq}(\mu,\nu)} \int_{X \times Z} \left[ \left( c(x, z) + \lambda_c \right) - \lambda_c \right] d\gamma(x, z) \tag{31}$$

where the cost function is mixed-sign and $\lambda_c = \max_{x,z}[-c(x, z)]$. It is worth noticing that $\lambda_m$ corresponds to $m$ which is user-defined, while $\lambda_c$ is self-determined by the cost structure. The reformulation provides insights into how elastic optimal transport is essentially different with partial optimal transport.

First, the fundamental difference is that ELOT preserves adaptive-mass while POT transports fixed-mass. For the POT problem, the goal of introducing the Lagrange multiplier $\lambda_m$ is to remove the fixed-mass constraint $\int_{X \times Z} d\gamma(x, z) = m$, making it easier to solve the POT problem. However, this does not eliminate the limitation that it needs to specify the mass budget $m$ (or equivalently find the appropriate value of the Lagrange multiplier $\lambda_m$), which is challenging because we usually have no prior knowledge on how much mass should be transported. For ELOT, we have no such a fixed-mass constraint, thus there is no need to introduce an extra Lagrange multiplier.

Second, ELOT determines the mass according to the task structure, while POT relies on the user to specify the mass budget. The reformulation gives some insights into how AOT attains adaptive-mass transport. According to Caffarelli & McCann (2010), for each mass $m$ there is a unique $\lambda$ corresponding to the $m$, and $m$ increases continuously as $\lambda$ is increased. For POT, the specific value of the Lagrange multiplier $\lambda_m$ is irrelevant to the task structure itself. On the contrary, ELOT self-determines the total mass, relying on the native structure the ground costs. Specifically, a definite $\lambda_c$ in ELOT results in a definite mass, while a larger $\lambda_c$ leads to the more mass to be transported.

Last but not least, ELOT provides a much larger capacity than POT by exploring the whole spectrum of mass instead of the fixed-mass. Consider the optimal transport problems with parameterized cost functions (Paty & Cuturi, 2019). Denote the parameterized cost function by $c_\theta(x, z)$ where $\theta$ is the learnable parameter. The elastic optimal transport with parameterized cost function can be formulated as

$$\min_{\substack{\theta, \\ \gamma \in \Gamma_{\leq}(\mu, \nu)}} \int_{X \times Z} c_\theta(x, z) d\gamma(x, z) \tag{32}$$

Likewise, it can be reformulated as

$$\min_{\substack{\theta, \\ \gamma \in \Gamma_{\leq}(\mu, \nu)}} \int_{X \times Z} \Big[ \big( c_\theta(x, z) + \lambda_{c_\theta} \big) - \lambda_{c_\theta} \Big] d\gamma(x, z) \tag{33}$$

where $\lambda_{c_\theta} = \max_{x, z}[-c_\theta(x, z)]$. The total transport mass of ELOT could increase continuously from 0 to 1 as $\lambda_{c_\theta}$ increases. Therefore, ELOT could attain the adaptive-mass ranged continuously across the whole spectrum of mass, thus offering a much larger capacity to search for the learnable parameters. In contrast, POT sticks to the user-specified mass budget no matter how the cost functions are varying, which is likely to be trapped into local optimums.

In summary, the distinctive advantages of ELOT against POT lie in three aspects: a) adaptive-mass preserving, b) self-determining according to task structure, c) larger capacity by exploring the spectrum of mass. Since the classical optimal transport with full mass constraints can be viewed the special case of partial optimal transport by setting $m = 1$, the claims hold for the classical optimal transport too.

### A.4 MORE EXPERIEMENT RESULTS

We report the implementation details, as well as more empirical evaluations.

#### A.4.1 IMPLEMENTATION DETAILS

For a fair comparison. we basically follow the settings of JUMBOT (Fatras et al., 2021) and m-POT (Nguyen et al., 2022) for the experiment setups.

**Networks.** Similar to (Fatras et al., 2021; Nguyen et al., 2022), we use ResNet-50 (He et al., 2016) pretrained on ImageNet as our extractor and one fully connected (FC) layer as our classifier for all datasets.

**Sampling.** We adopt class-balanced sampling on source domain where each class has the same number samples in a mini-batch, and random sampling is used on target domain since labels are unavailable in training, which is the same with (Fatras et al., 2021; Nguyen et al., 2022).

**Data Augmentation.** Following (Fatras et al., 2021; Nguyen et al., 2022), we use the same data pre-processing for all three datasets. The images are first resized into $256 \times 256$ and then randomly cropped with size of $224 \times 224$. Random translation/mirror and normalization are also applied for training. For testing, we adopt the ten-crop technique (Fatras et al., 2021; Nguyen et al., 2022) for robust results. Note that these settings are commonly used and the same as previous works (Fatras et al., 2021; Nguyen et al., 2022) for fair comparison.

**Training Details.** Following the settings of (Fatras et al., 2021; Nguyen et al., 2022), we adopt SGD optimizer with 0.9 momentum and $5e^{-4}$ weight decay for training, and the learning rates are set with the same strategy as (Ganin et al., 2016). Note that the learning rate of the classifier is set to be 10 times that of the extractor as the classifier is trained from scratch.

**Hyper-parameters.** For all three datasets, the weight of feature-wise cost $\alpha$ is set to 0.01. The weight of label-wise cost $\beta$ is set to 2.3, 9, and 4.5 for VisDA, Office-31, and Office-Home respectively. The entropy-regularized coefficient $\epsilon$ is set to 0.1, 1, and 1 for VisDA, Office-31, and Office-Home respectively. The batch size is set to 72, 62, and 65 for VisDA, Office-31, and Office-Home respectively, which are consistent with JUMBOT (Fatras et al., 2021) and m-POT (Nguyen et al., 2022). The experiments are trained for 2000, 5000, and 5000 iterations for VisDA, Office-31, and Office-Home, respectively.

### A.4.2  SENSITIVITY ANALYSIS

We report the sensitivity analysis with $\alpha$, $\beta$ and $\epsilon$ on VisDA dataset. Figure 2 shows the average accuracy and standard deviation varying with the parameters. It verifies the robustness of the proposed method on the hyper-parameters. For comparison, the best performance of m-POT (Nguyen et al., 2022) is also plotted as base line.

The coefficient $\alpha$ is used to adjust the weight of the feature-wise cost. The optimal range for $\alpha$ is between 0.004 and 0.012. ELOT yields the best accuracy on VisDA dataset when $\alpha = 0.01$. So we fix $\alpha = 0.01$ for the VisDA dataset in the experiments.

Note that $\beta$ is the weight to control the impact of label-wise cost. From Figure 2, we can see that ELOT attains the satisfactory performance when $\beta$ is greater than 2, and exhibits the high accuracies in a wide range of $\beta$.

The entropy-regularized coefficient $\epsilon$ is to control the sparsity of transport plan. It shows that entropy-regularized term helps getting better accuracies when $\epsilon$ increases from 0 to 0.1. The accuracy of ELOT reaches its maximum around $\epsilon = 0.1$. However when $\epsilon$ is too large (say $\epsilon = 1$), the accuracy falls. Nevertheless when $\epsilon$ varies in a wide spectrum from 0 to 0.5, ELOT beats the second best method m-POT (Nguyen et al., 2022).

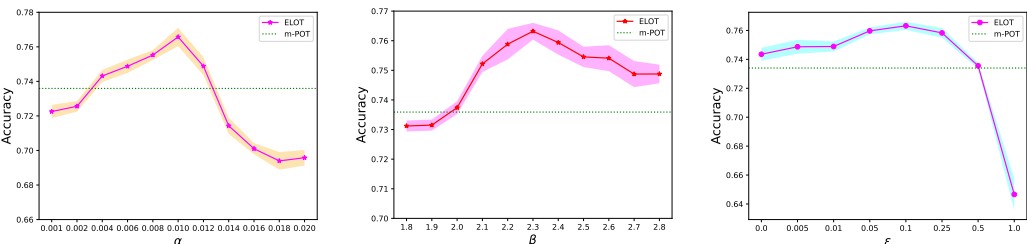

Figure 2: Sensitivity analysis with parameters $\alpha$, $\beta$, and $\epsilon$.

### A.4.3  T-SNE VISUALIZATION

We plot the T-SNE (Van der Maaten & Hinton, 2008) embeddings of four methods including Deep-JDOT (Damodaran et al., 2018), JUMBOT (Fatras et al., 2021), m-POT (Nguyen et al., 2022), and ELOT on VisDA dataset, in order to intuitively demonstrate the superiority of ELOT.

From Figure 3, we can see that the clusters of ELOT are more separate than those of the comparison methods. Meanwhile, the source samples and target samples with the same class label are relatively well aligned by ELOT. It suggests that our proposed method is able to better maintain the discriminative information in the course of domain alignment. It simultaneously aligns the domains and categories.

### A.5  RELATED WORK AND DISCUSSION

In real applications, the training and test data usually do not follow the independent and identically distributed assumption. Domain adaptation is the critical task in real-world machine learning applications since distribution discrepancy is ubiquitous in the data.

Most existing works on domain adaptation can be roughly classified into two categories: discrepancy-based methods and adversarial-learning based methods. The discrepancy-based methods explicitly minimized the domain distance using discrepancy metrics such as optimal transport distance or Maximum Mean Discrepancy (MMD) (Borgwardt et al., 2006). The domain adaptation methods based on optimal transport include OTDA (Courty et al., 2017b), ROT (Balaji et al., 2020), DeepJDOT (Damodaran et al., 2018), JUMBOT (Fatras et al., 2021), m-POT (Nguyen et al., 2022), etc. The typical methods based on MMD are JAN (Long et al., 2017b), PPDG (Tian et al., 2024), DeepONet (Goswami et al., 2022), to name a few. The adversarial-learning based methods aim to learn domain-invariant representations via adversarial training. The typical methods include DAN-

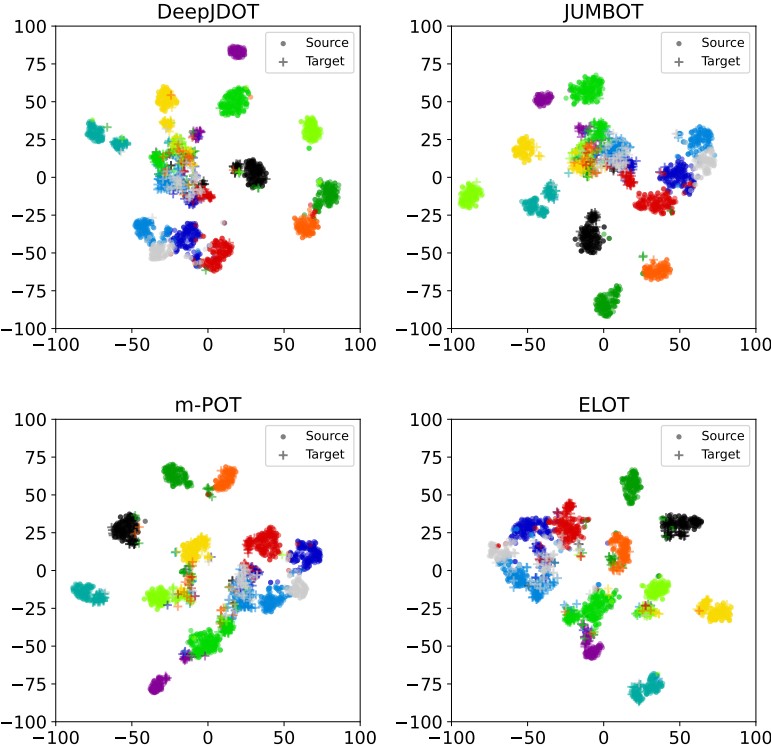

Figure 3: T-SNE embeddings of 2000 samples of VisDA for different methods. Samples are colored based on their class labels.

N (Ganin et al., 2016), CDAN (Long et al., 2017a), ALDA (Chen et al., 2020), DrugBAN (Bai et al., 2023), CityTrans (Ouyang et al., 2024), etc. Please refer to the survey paper (Wilson & Cook, 2020) for more details.

We demonstrate the effectiveness of the instantiation of ELOT in domain adaptation tasks. However, it is widely applicable to various branches of machine learning due to the fact that the training and text data in real applications usually do not follow the independent and identically distributed assumption.

Furthermore, although we focus on machine learning applications here, it is worth noting the applications in a wide variety of areas beyond artificial intelligence can opt for elastic optimal transport whenever the adaptive-mass preserving is preferred for the alignment of probability distributions. This is indeed the case since full-mass or fixed-mass preserving is too restrictive in many scenarios.

