# OpenReview forum: "Elastic Optimal Transport: Theory, Application, and Empirical Evaluation"
_ICLR.cc/2026/Conference — ICLR 2026 Poster_

### Official Review · Reviewer_qzrK · 2025-10-28

**Soundness:** 3
**Presentation:** 3
**Contribution:** 3
**Rating:** 6
**Confidence:** 4

**Summary:**

This paper considers the mathematical aspects of optimal transport (OT), where a variant of existing OT formulation is proposed. Specifically, the authors focus on the strict equality or inequality constraints on transport mass, and try to propose a new mechanism to achieve a relaxation on the strict hard constraints, i.e., adaptive mass. Empirical validations are conducted on standard domain adaptation datasets compared with other OT counterparts.

**Strengths:**

S1. A new OT formulation with elastic constraints and theoretical analysis.

S2. The organization is clear and the writing is easy to follow.

S3. The experimental performance is superior to other standard OT variants.

**Weaknesses:**

W1. The rigor of the claim and theoretical analysis should be improved.

W2. The empirical validation could be improved, where the SOTA comparison methods should be considered.

**Questions:**

Q1. The elastic optimal transport (ELOT) is not rigorously defined and analyzed. Specifically, the notation $\mathbb{R}_{±}$ in Eq. (6) for the cost function is not defined. If it implies that the cost function could be both positive and negative, the ELOT could not be well-defined, e.g., does it still satisfy the metric property? A rigorous and complete theoretical analysis is necessary to ensure the validity of ELOT.

Q2. The ELOT seems to be an incremental work of partial OT. Specifically, the basic conclusions and technical details (i.e., Thm. 1 and its proofs) of ELOT are almost the same as partial OT [27, Prop. 1], while the proofs in line 54-56 manuscript seem to be problematic, i.e., there are no rigorous proofs to ensure that the submatrix of augmented OT plan is equivalent to the OT plan of ELOT.

Q3. Moreover, the formulation of ELOT in Eq. (6)-(7) still cannot ensure the adaptive learning of mass value $s$. Specifically, the technique based on dummy variables still induces the parameter $\sigma$ in Eq. (8), which is basically equivalent to adjusting $s$.

Q4. In the claim below Eq. (8), the authors set $\sigma$ as 0 in the experiment. However, in such a formulation, the ELOT problem seems to be exactly the original Kantorovich OT problem, as the augmented cost matrix is block-diagonal where the $n\times m$ block and $1\times 1$ block are independent. Based on this problem, it would be hard to understand the performance improvement of ELOT in experiments.

Q5. The comparison experiment with SOTA methods is not significant. This experiment is conducted on 3 DA datasets, where the performance is significantly lower than the SOTA methods in recent years. Some further verifications on the latest proposed datasets and advanced baselines are necessary, e.g., DomainNet dataset and ViT backbone.

Q6. The quality of OT plan learned by ELOT is not sufficiently validated. For example, if ELOT can adaptively adjust the mass of plan, a quantitative/qualitative comparison between the OT plans of different OT formulations should be considered.

---

> ### Author Response · Authors · 2025-11-21
> **Response to Reviewer qzrK**
>
> Thank you very much for constructive comments. The point-to-point responses are given as follows.
>
> **C15:**
> rigor of statement.
>
> **A15:**
> Thanks. We would like to clarify that we had defined $\mathbb{R}_{\pmb{\pm}}$ as a mixed-sign matrix in the preliminary section. Although the cost with mixed-sign items is no longer the distance metric, the combination of mixed-sign costs and marginal inequality constraints brings with significant benefit by enabling adaptive-mass transportation, leading to better generalization performance. We conducted complete theoretical analysis to ensure the validity of ELOT in the theoretical section, which provided insights into the solution structure and the mass transportation mechanism of ELOT.
>
> ****
>
> **C16:**
> Relation with partial OT.
>
> **A16:**
> First, we would like to clarify that ELOT is fundamentally distinctive from partial OT. Please see our response (\#A2) to all the reviewers above for the details.
>
> Also, we feel there are typos in the comment “the proofs in line 54-56” since line 54-56 is irrelevant to the proof. The rigorous proofs have been given to ensure that the submatrix of augmented OT plan is equivalent to the OT plan of ELOT.
>
> ****
>
> **C17:**
> adaptive learning of mass value.
>
> **A17:**
> We would like to clarify that although the technique based on dummy variables induces $\sigma$, the solutions of ELOT is irrelevant to the specific value of $\sigma$, as indicated in Theorem 1. Therefore, choosing the value of $\sigma$ is irrelevant to adjust the mass $s$. ELOT relies on the combination of mixed-sign cost and the marginal inequality constraints for adaptive-mass learning.
>
> ****
>
> **C18:**
> Relation with Kantorovich OT.
>
> **A18:**
> ELOT is distinctive from Kantorovich OT since their problem formulations are different. As indicated in Section 2.3, the solution structure and the mass transportation mechanism of ELOT are fundamentally distinctive from Kantorovich OT.
>
> ****
>
> **C19:**
> SOTA method comparison.
>
> **A19:**
> Thanks. We include the DomainNet dataset and the recent SOTA methods such as STCPDA [1] and ARPM [2] in the empirical evaluations. DomainNet is a large-scale challenging dataset with 345 classes.  We follow ARPM [2] to use the first 40 classes in alphabetical order to build the target domain in each task. Table 1 reports the average classification accuracy on target domain for the six partial domain adapatation tasks. ELOT significantly outperforms both STCPDA [1] and ARPM [2], demonstrating its superiority on the large-scale challenging dataset.
>
> **Table 1: Classification accuracy on DomainNet for *partial* domain adaptation tasks.**
> | Method | C $\rightarrow$ P | C $\rightarrow$ R | C $\rightarrow$ S | S $\rightarrow$ C | S $\rightarrow$ P | S $\rightarrow$ R | Avg |
> | :--- | :---: | :---: | :---: | :---: | :---: | :---: |:---: |
> | STCPDA [1] | 65.1 | 69.6 | **69.6** | 64.4 | 60.7 | 67.8 | 66.2 |
> | ARPM [2] | 67.9 | **79.8** | 66.3 | 62.5 | 64.8 | 71.7 | 68.8 |
> | **ELOT (ours)** | **68.8** | 77.7 | 68.3 | **68.0** | **69.9** | **75.9** | **71.4** |
>
> [1] Addressing the overfitting in partial domain adaptation with self-training and contrastive learning. IEEE Transactions on Circuits and Systems for Video Technology. 34(3): 1532-1545 (2024).
> [2] Adversarial reweighting with $\alpha$-power maximization for domain adaptation. International Journal of Computer Vision 132(10): 4768-4791 (2024).
>
> ****
>
> **C20:**
> OT plan.
>
> **A20:**
> We will include more empirical evaluations of OT plan on synthetic dataset in order to intuitively verify the superiority of ELOT over the other OT formulations.

---

### Official Review · Reviewer_B4zg · 2025-10-30

**Soundness:** 2
**Presentation:** 3
**Contribution:** 2
**Rating:** 4
**Confidence:** 5

**Summary:**

The paper proposes Elastic Optimal Transport (ELOT), an optimal transport (OT) formulation designed to address practical limitations of classical OT, partial OT, and unbalanced OT. In standard OT, all mass from the source distribution must be transported to the target distribution. Partial OT transports only a fixed budget $s$ of mass, which the user must choose. Unbalanced OT relaxes the marginal constraints but requires tuning divergence penalties. ELOT instead allows the optimal plan to decide how much mass, depending only on the cost. ELOT explicitly allows the cost matrix $C$ to include negative entries, and thus adaptively determines the transported mass. The authors apply ELOT to domain adaptation and compare it with OT-based approaches.

**Strengths:**

- The motivation is strong. The authors aim to adaptively match datapoints without hand-tuning transport mass, which is really useful for practical applications if realized.
- The authors consider a signed cost to realize the adaptive amount of transported mass, which is reasonable.
- The computation is transformed into an OT formulation, and thus can be solved by existing tools.
- The paper is easy to read.

**Weaknesses:**

- Novelty relative to partial OT / unbalanced/robust OT needs to be strengthened.  Partial OT already allows transporting only part of the mass. In classical formulations, one can introduce a Lagrange multiplier that effectively shifts the cost matrix and induces an “automatic” choice of how much mass to transport. Prior work (the paper cites Caffarelli & McCann, 2010) can create a situation where negative effective cost encourages matching only for “good pairs,” and the transported mass adapts as a function of that shift.
ELOT is described as more general because it directly allows mixed-sign costs and only imposes marginal $\le$ constraints, and it claims to “automatically finds the optimal mass to be transferred without setting a priori budget.” However, ELOT can still look like “partial OT with a specific cost shaping and slack embedding.” The paper should give a crisper mathematical argument for why ELOT is fundamentally different and not just a repackaging.

- Calling $W(\mu,\nu)$ a ‘distance’ may be misleading. Because $C$ can be non-symmetric and can contain negative values, the induced objective $W(\mu,\nu)$ is not guaranteed to be a metric or even a divergence: non-negativity, symmetry, and triangle inequality can fail.
The paper uses “optimal transport distance” language, which is standard OT terminology, but here it risks overstating the geometric meaning. This should be toned down or clarified. Meanwhile, the commonly used costs, e.g., L1 L2, are not suitable. How could ELOT be applied to partial transport under such widely utilized costs?

- The paper aims to avoid hand-tuning transport mass, a hyperparameter. However, the costs in the domain adaptation task and the partial domain adaptation task involve hyperparameters, to which the performance is sensitive, as shown in the experiments. So I feel like the authors avoid tuning one parameter but instead tune several other parameters. For DA, the parameters are set to the values in the compared methods. But for other applications, how these parameters are determined.

**Questions:**

- Could the authors explain how to define the cost for a general task, as the commonly utilized costs are often non-negative in most tasks?
- Can the private class data be detected and omitted by ELOT in partial DA?
- The compared methods are not soTA. Could the authors include more recent SOTA? For example, MOT[1], ARPM[2], both are OT-based.

[1] Mot: Masked optimal transport for partial domain adaptation

[2]Adversarial Reweighting with α-Power Maximization for Domain Adaptation

---

> ### Author Response · Authors · 2025-11-21
> **Responses to Reviewer B4zg**
>
> Thank you very much for constructive comments. The point-to-point responses are given as follows.
>
> **C10:**
> novelty relative to partial OT.
>
> **A10:**
> Please see our response (\#A2) to all the reviewers above.
>
> ****
>
> **C11:**
> distance metric.
>
> **A11:**
> Thanks. Regarding the mixed-sign cost, although it is no longer a distance metric, it facilitates the adaptive-mass transport as the return in our ELOT formulation.  This could be a limitation in some tasks, but ELOT is widely applicable to many scenarios such as domain adaptation where the cost could be induced in multiple feature spaces (e.g., original feature space and latent feature space, or marginal distribution and conditional distribution).
> Meanwhile, although the L1/L2-norm cost is not directly suitable, their linear combination is still applicable.
>
> ****
>
> **C12:**
> tuning transport mass, and general cost.
>
> **A12:**
> Thanks. We would like to clarify that we use exactly the same number of hyper-parameters as many other prominent OT-based methods such as unbalanced OT  (e.g., JUMBOT) and partial OT (e.g., m-POT) to construct the cost function. Since one may move the coefficient $\alpha$ from the cost term to the cross-entropy loss term as usually done to achieve the same effect, there is actually only one hyper-parameter $\beta$ used in the cost function. Besides, both JUMBOT and m-POT need extra trade-off parameters ($s, \tau_1, \tau_2$ ) to set either the marginal penalization coefficient or a priori amount of mass to be transferred, which are not needed in ELOT. Therefore, ELOT uses less hyper-parameters than many other prominent OT-based methods.
>
> Define the cost in both original feature space $g(\cdot)$ and the latent feature spaces $f(\cdot)$ is a common practice, which could lead to better generalization performance. This kind of cost function is widely applicable to a general task beyond domain adaptation.
> \begin{equation}
> \mathcal{C}_{ij} = \big\|\big\| g(x_i) - g(z_j) \big\|\big\|^2 + \beta \big\|\big\| f(x_i) - f(z_j) \big\|\big\|^2.
> \end{equation}
>
> Instead, we could use a slightly different one to obtain the mixed-sign cost.
> \begin{equation}
> \mathcal{C}_{ij} = \big\|\big\| g(x_i) - g(z_j) \big\|\big\|^2 - \beta \big\langle f(x_i),  f(z_j) \big\rangle.
> \end{equation}
> It is unavoidable to tune the hyper-parameter $\beta$ in these cases since this hyper-parameter is a part of the cost structure, depending on the problem itself. As usually, it can be tuned via cross-validation.
>
> Therefore, we would like to clarify that ELOT relies on the cost structure for adaptive-mass transport, rather than playing the exchange game of hyper-parameter.
>
> ****
>
> **C13:**
> detect private class.
>
> **A13:**
> ELOT provides a robust solution to align the probability distributions adaptively by respecting the outliers and divergences in data. Therefore, ELOT has potentials to detect the private classes. It is one of the many interesting research directions along with ELOT.
>
> ****
>
> **C14:**
> more recent SOTA.
>
> **A14:**
> Thanks. We include the DomainNet dataset and the recent SOTA methods such as STCPDA [1] and ARPM [2] in the empirical evaluations. DomainNet is a large-scale challenging dataset with 345 classes.  We follow ARPM [2] to use the first 40 classes in alphabetical order to build the target domain in each task. Table 1 reports the average classification accuracy on target domain for the six partial domain adapatation tasks. ELOT significantly outperforms both STCPDA [1] and ARPM [2], demonstrating its superiority on the large-scale challenging dataset.
>
> **Table 1: Classification accuracy on DomainNet for *partial* domain adaptation tasks.**
> | Method | C $\rightarrow$ P | C $\rightarrow$ R | C $\rightarrow$ S | S $\rightarrow$ C | S $\rightarrow$ P | S $\rightarrow$ R | Avg |
> | :--- | :---: | :---: | :---: | :---: | :---: | :---: |:---: |
> | STCPDA [1] | 65.1 | 69.6 | **69.6** | 64.4 | 60.7 | 67.8 | 66.2 |
> | ARPM [2] | 67.9 | **79.8** | 66.3 | 62.5 | 64.8 | 71.7 | 68.8 |
> | **ELOT (ours)** | **68.8** | 77.7 | 68.3 | **68.0** | **69.9** | **75.9** | **71.4** |
>
> [1] Addressing the overfitting in partial domain adaptation with self-training and contrastive learning. IEEE Transactions on Circuits and Systems for Video Technology. 34(3): 1532-1545 (2024).
> [2] Adversarial reweighting with $\alpha$-power maximization for domain adaptation. International Journal of Computer Vision 132(10): 4768-4791 (2024).

---

### Official Review · Reviewer_y8JA · 2025-10-31

**Soundness:** 4
**Presentation:** 3
**Contribution:** 3
**Rating:** 6
**Confidence:** 4

**Summary:**

The paper introduces Elastic Optimal Transport (ELOT), a novel formulation of optimal transport that relaxes the full-mass or fixed-mass constraints present in classical OT, partial OT, and unbalanced OT. ELOT allows for adaptive-mass preservation and supports mixed-sign cost matrices, making it more flexible for real-world applications where noise, outliers, or distribution shifts are present. The authors provide theoretical analysis, an equivalent reformulation solvable with standard OT solvers, and apply ELOT to unsupervised and partial domain adaptation tasks. Experimental results on standard benchmarks (VisDA, Office-31, Office-Home) demonstrate that ELOT outperforms several state-of-the-art OT and non-OT baselines.

**Strengths:**

1.	Novel Formulation: The idea of adaptive-mass transport is well-motivated and addresses a clear limitation of existing OT methods. The introduction of a mixed-sign cost matrix enhances its applicability to real-world problems.
2.	Strong Empirical Performance: The paper provides extensive and convincing experiments on multiple domain adaptation benchmarks, showing consistent and significant improvements over a range of strong baselines.
3.	Theoretical-Practical Bridge: The work offers valuable theoretical insights and a practical reformulation that enables the use of standard OT solvers, facilitating wider adoption.

**Weaknesses:**

Proofs of Theorems 1 and 2 Lack Rigor:
The proofs of Theorems 1 and 2, while intuitively appealing, lack rigor in their current form. In particular, the argument that two transport plans are equivalent because they minimize the same cost ignores the issue of non-uniqueness of solutions in ELOT—unlike in classical OT where uniqueness often holds under mild conditions. The authors should revisit the proofs to account for potential multiple optima. The theorems could be rephrased to state that there exists an optimal plan for the reformulated problem that matches an optimal plan of the original ELOT, rather than implying equality of all such plans.

**Questions:**

1.  Potential Degeneracy When Transport Mass is Zero:
The formulation of ELOT allows for the possibility that the total transported mass s=0, which occurs when all ground costs are positive. While this may be desirable in some outlier-rich scenarios, it could also lead to degenerate solutions in applications where some degree of alignment is necessary.
Suggestion: The authors should discuss this property and its implications for the general applicability of ELOT. A discussion on whether this poses a limitation in practice and if potential modifications or safeguards could mitigate this issue would strengthen the paper.
2. Ambiguity in Theoretical Relationship Between ELOT and Unbalanced Optimal Transport:
In the current formulation, ELOT is presented as distinct from unbalanced OT. However, in unbalanced OT, the marginal divergence penalties act as soft constraints. when τ1,τ2→0 the penalties vanish, allowing relaxed inequalities γ1_m ≤μ,γ^T 〖1〗_n≤ν —which closely resembles the ELOT formulation. The paper currently does not analyze this limiting behavior, making it unclear whether ELOT can be regarded as a limiting case of unbalanced OT.
Suggestion:
To further strengthen the theoretical contribution, the authors may consider discussing the limiting behavior of unbalanced OT as the penalty coefficients approach zero, and clarifying how ELOT relates to this case. This would help position ELOT more clearly within the broader OT framework.

---

> ### Author Response · Authors · 2025-11-21
> **Responses to Reviewer y8JA**
>
> Thank you very much for constructive comments. The point-to-point responses are given as follows.
>
> **C7:**
> rigor of statements.
>
> **A7:**
> Thanks. We would like to clarify that either ELOT or classical optimal transport does not guarantee a unique solution, which stems from the interplay between its linear objective function and the convex polyhedral feasible region. Therefore, both the original ELOT and the reformulated problem could have multiple optima. However, the uniqueness of solution to the entropy-regularized ELOT and its reformulated counterpart is guaranteed because the entropy-regularized objective function is strictly convex.
> We add the discussion in the revised manuscript to enhance the theoretical analysis.
>
> ****
>
> **C8:**
> potential degeneracy.
>
> **A8:**
> Thank you for the considerate suggestion.
> We would like to clarify that ELOT requires the costs to be mixed-sign to avoid the potential degeneracy. This could be a limitation in some tasks, but ELOT is widely applicable to many scenarios such as domain adaptation where the cost could be induced in multiple feature spaces (e.g., original feature space and latent feature space, or marginal distribution and conditional distribution).
>
> In the future, we will develop a universal formulation of optimal transport. The basic idea is to unify optimal transport with various constraints (full-mass, fixed-mass, and adaptive-mass) into one universal formulation which is cost-aware. It can automatically turn into the specific formulation of optimal transport, depending on the structure of costs.
> - If all the costs are positive, it becomes partial optimal transport.
> - If all the costs are negative, it becomes Kantorovich optimal transport.
> - If there co-exist positive and negative costs, it becomes elastic optimal transport.
>
> ****
>
> **C9:**
> relation with unbalanced optimal transport.
>
> **A9:**
> Please see our response (\#A1) to all the reviewers above.

---

### Official Review · Reviewer_sLP5 · 2025-10-31

**Soundness:** 3
**Presentation:** 3
**Contribution:** 2
**Rating:** 4
**Confidence:** 4

**Summary:**

This paper proposes Elastic Optimal Transport (ELOT), a novel formulation using marginal inequalities and a mixed-sign ground cost. The authors present two key theoretical results: an equivalence to an equality-constrained problem whose plan is invariant to a parameter σ (Thm. 1), and a "mass transport mechanism" (Thm. 2) showing that mass flows only on negative-cost entries, which enables automatic outlier filtering. Empirically, the method demonstrates consistent improvements over domain adaptation baselines on VisDA, Office-31, and Office-Home.

**Strengths:**

In general the method is interesting and promising, avoiding limitations of the current OT solvers. Theoretically justified formulation with consistent gains in UDA and partial DA using a uniform backbone and setup.

**Weaknesses:**

While the formulation is clean and intuitive, my primary concern is the unclear formal relationship to Unbalanced Optimal Transport (UOT). The authors mentioned UOT in the background but did not consider these methods in more detail. Although UOT is a well-known approach to a similar problem to that stated in the paper, a more detailed analysis of its relationship to existing solvers is necessary. The  The authors assert a connection, but do not analyse it sufficiently. Experimental comparison to UOT is completely ignored. Why? To strengthen the contribution, please consider addressing the following questions:

**Questions:**

**Q1**: Under what specific assumptions on the cost C or marginals (μ,ν) do ELOT and UOT provably yield different couplings?

**Q2**: Does elastic solver can be considered in entropy-based regularized settings? Given that negative costs can cause instability in entropic solvers (e.g., Sinkhorn's exp(−C/ε)), what stabilization techniques are used? Does the plan invariance (Thm. 1) hold exactly under this entropic regularization?

**Q3**: The paper defines the resulted W and calls it an “optimal transport distance,” but fails to discuss its metric properties as identity, symmetry. Does the resulted solution value actually provide some sort of Wasserstein distance?  I the cost matrix C has no negative entries, the zero plan (γ=0) is feasible and optimal, yielding W(μ,ν)=0 by construction. This seems to violate the identity property of a distance and is consistent with Theorem 2, which states mass only flows on negative-cost entries.

minors:
Add runtime/memory measurements or complexity analysis to justify statements.

---

> ### Author Response · Authors · 2025-11-21
> **Responses to Reviewer sLP5**
>
> Thank you very much for constructive comments. The point-to-point responses are given as follows.
>
> **C3:**
> primary concern is the unclear formal relationship to Unbalanced Optimal Transport (UOT)...do ELOT and UOT provably yield different couplings?
>
> **A3:**
> Please see our response (\#A1) to all the reviewers above.
>
> ****
>
> **C4:**
> entropy-based regularized settings.
>
> **A4:**
> Thanks. Elastic solver can be considered in entropy-based regularized settings, as stated in Section 3. For the instability issue in entropic solvers, one practical stabilization technique is to normalize the cost via $C' = \frac{C}{M}$, where $M$ is a sufficiently large constant such as $M= \max_{i,j}[-C_{ij}]$.  .
> The solution of transport plan is determined by the general theory of linear programming and is independent of the sign of the cost. Therefore, the invariance of transport plan still holds under this entropic regularization.
>
> ****
>
> **C5:**
> distance metric.
>
> **A5:**
> Thanks. We will discuss the metric property in more details. Regarding the mixed-sign cost, although it is no longer a distance metric, it facilitates the adaptive-mass transport as the return in our ELOT formulation. Also, we would like to clarify that ELOT requires the costs to be mixed-sign to avoid the potential degeneracy. This could be a limitation in some tasks, but ELOT is widely applicable to many scenarios such as domain adaptation, where the cost could be induced in multiple feature spaces (e.g., original feature space and latent feature space, or marginal distribution and conditional distribution).
>
> ****
>
> **C6:**
> complexity analysis.
>
> **A6:**
> The algorithm complexity of ELOT is slightly different from classical optimal transport with the scale of problem-solving changing from $n \times m$ to $(n+1) \times (m+1)$, where the extra one corresponds to the augmented dimension.

---

### Author Response · Authors · 2025-11-21
**Responses to all reviewers (part 1 of 2)**

We deeply appreciate your insightful comments and suggestions. The responses to the common concerns about the relations between elastic optimal transport (ELOT) and classical optimal transport are given as follows. We clarify that ELOT is fundamentally different from unbalanced optimal transport and partial optimal transport, highlighting the novelty of ELOT.

**C1: ELOT vs unbalanced optimal transport (Reviewers sLP5, y8JA)**

**A1:**
We would like to clarify that ELOT is NOT a limiting case of unbalanced optimal transport (UOT).  As the penalty coefficients approach zero, i.e., $\tau_1 \rightarrow 0$ and $\tau_2 \rightarrow 0$, allowing relaxed inequalities, UOT will lead to the degenerate solution with zero transport plan due to the nonnegative constraint of costs. Even though UOT is allowed to accept the mixed-sign cost (which means the nonnegative constraint on costs is violated), it will lead to the solution with infinite mass flowing on negative-cost entries due to the overly relaxed inequalities $\tau_1 D_{\phi}(\gamma 1_m \| \mu)  + \tau_2 D_{\phi}(\gamma^{T} 1_n \| \nu) $ in the case of $\tau_1 \rightarrow 0$ and $\tau_2 \rightarrow 0$. In contrast, ELOT depends on the combination of the marginal inequality constraints and mixed-sign cost to achieve adaptive-mass transport.

Therefore, ELOT is certainly not the limiting case of unbalanced OT. Instead, ELOT is fundamentally distinctive from unbalanced optimal transport. First, the problem formulations are different. Second, the mass transportation mechanism is different, as indicated the theoretical analysis. ELOT can achieve adaptive-mass transport, relying on the combination of the marginal inequality constraints and mixed-sign cost, while UOT depends on users to specify the hyper-parameters of soft constraints.

Also, we would like to clarify that the typical UOT method called JUMBOT was already included in the empirical comparison, and ELOT significantly outperforms JUMBOT, as reported in the experiment section.

---

### Author Response · Authors · 2025-11-21
**Responses to all reviewers (part 2 of 2)**

**C2: ELOT vs partial optimal transport (Reviewers B4zg, qzrK)**

**A2:**
We go deep into the relation and difference between ELOT and partial optimal transport (POT), and highlight the distinctive advantages of ELOT against POT.

Without loss generalization, let's assume both $\mu$ and $\nu$ are probability measures ($\mu[X]=\nu[Z]=1$) here for simplicity. Partial optimal transport [1] is formulated as
\begin{equation}
\mathop{\min}\limits_{\gamma \in \Gamma_{\le}(\mu,\nu), \atop \gamma[X \times Z] = m} \int_{X \times Z} c^+(x, z) d \gamma(x,z)
\end{equation}
where the cost function $c^+(x, z)$ is non-negative. Caffarelli and McCann introduced a Lagrange multiplier $\lambda_m \ge 0$ conjugate to the fixed-mass constraint $\gamma[X \times Z] = m$ and reformulated the POT problem as
\begin{equation}
\mathop{\min}\limits_{\gamma \in \Gamma_{\le}(\mu,\nu)} \int_{X \times Z} \big[ c^+(x, z) - \lambda_m \big] d \gamma(x,z).
\end{equation}
Likewise, ELOT can be reformulated as
\begin{equation}
\mathop{\min}\limits_{\gamma \in \Gamma_{\le}(\mu,\nu)} \int_{X \times Z} \Big[ \big(c(x, z) + \lambda_c \big) - \lambda_c \Big] d \gamma(x,z)
\end{equation}
where the cost function is mixed-sign and $\lambda_c = \mathop{\max}_{x,z}[-c(x,z)]$. The reformulation provides insights into how ELOT is essentially different from partial optimal transport.

First, the fundamental difference is that ELOT preserves adaptive-mass while POT transports fixed-mass. For the POT problem, the goal of introducing the Lagrange multiplier $\lambda_m$ is to remove the  fixed-mass constraint $\gamma[X \times Z] = m$, making it easier to solve the POT problem. However, this does not eliminate the limitation that it needs to specify the mass budget $m$ (or equivalently find the appropriate value of the Lagrange multiplier $\lambda_m$), which is challenging because we usually have no prior knowledge on how much mass should be transported. For ELOT, we have no such a fixed-mass constraint, thus there is no need to introduce an extra Lagrange multiplier.

Second, ELOT determines the mass according to the task structure, while POT relies on the user to specify the mass budget. The reformulation gives some insights into how ELOT attains adaptive-mass transport. According to [1], for each mass $m$ there is a unique $\lambda$ corresponding to the $m$, and $m$ increases continuously as $\lambda$ is increased. For POT, the specific value of the Lagrange multiplier $\lambda_m$ is irrelevant to the task structure itself. On the contrary, ELOT self-determines the total mass, relying on the native structure the ground costs.  Specifically, a definite $\lambda_c$ in ELOT results in a definite mass, while a larger $\lambda_c$ leads to the more mass to be transported.

Last but not least, ELOT provides a much larger capacity than POT by exploring the whole spectrum of mass instead of the fixed-mass. Consider optimal transport problems with parameterized cost functions. Denote the parameterized cost function by $c_{\theta}(x,z)$ where $\theta$ is the learnable parameter. The adaptive optimal transport with parameterized cost function can be formulated as
\begin{equation}
\mathop{\min}\limits_{\theta, \atop \gamma \in \Gamma_{\le}(\mu,\nu)} \int_{X \times Z} c_{\theta}(x, z) d \gamma(x,z)
\end{equation}
Likewise, it can be reformulated as
\begin{equation}
\mathop{\min}\limits_{\theta, \atop \gamma \in \Gamma_{\le}(\mu,\nu)} \int_{X \times Z} \Big[ \big(c_{\theta}(x, z) + \lambda_{c_{\theta}} \big) - \lambda_{c_{\theta}} \Big] d \gamma(x,z)
\end{equation}
where  $\lambda_{c_{\theta}} = \max_{x,z}[-c_{\theta}(x,z)]$. The total transport mass of ELOT could increase continuously from 0 to 1 as $\lambda_{c_{\theta}}$ increases. Therefore, ELOT could attain the adaptive-mass ranged continuously across the whole spectrum of mass, thus offering a much larger capacity to search for the learnable parameters. In contrast, POT adheres to the user-specified mass budget no matter how the cost functions vary.

In summary, the distinctive advantages of ELOT against POT lie in three aspects: a) adaptive-mass preserving, b) self-determining according to task structure, c) larger capacity by exploring the spectrum of mass.  Since the classical optimal transport with full mass constraints can be viewed the special case of partial optimal transport by setting $m=1$, the claims hold for the classical optimal transport too.

[1] Luis Caffarelli and Robert J. McCann. Free boundaries in optimal transport and Monge-Ampere obstacle  problems. Annals of Mathematics, 171:673--730, 2010.

---

### Comment · Area_Chair_2t4Z · 2025-11-25
**Discussions started**

Dear reviewers,

The authors have provided responses to the reviews. Please read these responses and discuss with authors if you have any further comments/questions.

Best,

AC

---

### Author Response · Authors · 2025-12-01
**Summary of Changes in the Revised Manuscript**

We would like to express our sincere gratitude to all the reviewers again. Based on the comments, we made substantial improvements to the manuscript. Here is the summary of the major changes in the revised PDF:

- We clarify that ELOT is fundamentally different from unbalanced optimal transport and partial optimal transport, in order to highlight the novelty of ELOT and further enhance the theoretical analysis. Please see the new added section (Section A.3 Relations with Existing Optimal Transport) for details.

- We include the DomainNet dataset and more recent SOTA methods to enhance the empirical evaluations. The results reinforce the superiority of ELOT on large-scale challenging datasets. Please see Section 4.3 for details.

- We add the discussion of entropic ELOT and the uniqueness of solution to ELOT to enhance the theoretical analysis. Please refer to Section 2.2 for details.

- We discuss the limitation of mixed-sign cost in some tasks and its wide applicability to general scenarios where the cost could be induced in various feature spaces. Please refer to Section 3 for details.

- We add the discussion of algorithm complexity in Section 2.2.

We believe that all the concerns should have been addressed, and hope the new version of manuscript could meet the expectation of all reviewers.

---

### Meta-Review · Area_Chair_cDgx · 2026-01-02

**Summary:**

This paper presents a novel Optimal Transport (OT) formulation: Elastic Optimal Transport (ELOT). This formulation relaxes the full-mass or fixed-mass constraints that are considered in classic OT, partial OT and unbalanced OT. ELOT allows to determine the adaptively the transport mass depending on the cost, this is achieved by allowing the use of mixed-sign cost matrices. The method is evaluated on Domain Adaptation tasks.

Reviewer sLP5 mentions that the method is interesting and promising, avoiding current limitations with theoretical justifications. The main concern of the reviewer lies in the lack of detailed analysis and comparison with Unbalanced Optimal Transport. Other minor questions concern entropic regularization, the notion of distance metric and complexity analysis.

Reviewer y8JA underlines on the positive side the novelty of the formulation, strong empirical performance and theoretical support. On the other hand, the reviewer identifies that the proofs of the main theorems lack rigor and the theorems should be rephrased. He also mentions a case of potential degeneracy and that the link with Unbalanced OT is insufficiently done.

Reviewer B4zg identifies on the positive side the quality of the motivation, the use of a signed cost for adaptive transport mass, the fact that the paper is easy to read. On the other hand, the reviewer mentions weaknesses that include the lack of justification of the novelty with respect to other OT approaches, the use of the term distance that can be misleading, and has an issue with parameter tuning in applications. He also asked questions on the definition of a cost for a general task, the private class data in partial DA and the lack of SOTA methods in the experiments.

Reviewer qzrK mentions on the positive side a novel OT method with theoretical justification, clear organization, experimental results superior to standard OT variants. Among the weaknesses, he underlines the lack of rigor of the claims, the theoretical analysis that can be improved and empirical evaluation that can be improved with other SOTA methods.

Overall, the paper proposes a novel OT approach that allows more flexibility by relaxing the usual mass constraints. Authors have justified the novelty with respect to other methods. Empirical results show improvements with respect to compared baselines on DA tasks. The evaluation could have considered more recent SOTA methods (notably for UOT). Presentation of the theoretical results could be strengthened.
The contribution is interesting with some drawbacks.
Can be accepted if there is enough room.

**Reviewer Concerns:**

An important concern of the paper was the positioning with respect to unbalanced optimal transport (mainly reviewers sLP5 and y8JA) and partial optimal transport (B4zg, qrzK). Authors have provided detailed justifications for justifying that ELOT is different and brings novelty with respect to the other 2 OT methods. A new section has been added in the Appendix of the revised version of the paper. I guess that this part answers this important issue.

About the experiments, authors have added a new dataset (DomainNet) and two new recent methods on partial optimal transport of 2024 (including one of the two proposed by reviewer B4zg). I am nevertheless concerned by the fact that the standard deviations of each experiment are not indicated for all the baselines (except in Table 2). This answers in part some concerns of reviewers qzrK and B4zg. For unbalanced transport, only one method is considered in the experiments (if I'm correct), so the comparison is still limited and does not answer fully the concern of reviewer sLP5.

Some reviewers had concerns with the lack of rigor. Authors have added some precisions in Section 2.2, but the improvement seems limited, it is difficult to know if the addition is sufficient without the feedback of the reviewers.

Authors have added some precisions on the limitation of the method, the algorithm complexity and other precisions.

The new revision includes all the elements mentioned above, which improves clearly the paper. The last point subject to discussion is the significance of the method: empirical results report good performance, but the setup contains a limited number of recent methods.

**Reviewer Scores:**

y8JA gave a 6. He had many reservations about the lack of rigor and proofs. The proofs did not change much, authors have added some precisions. I do not feel that the reviewer would have increased his score.

B4zg gave a 4. The novelty with respect to partial OT, UOT and robust OT has been addressed by authors I guess. There has been another remark on tuning and I am not sure that the reviewer would be satisfied by answer. I think that the reviewer could have increased his score to 5.

sLP5 gave a 4. In my opinion have positively answers to the concern related to the link between ELOT and UOT, but the experimental evolution did not include much and recent methods on UOT. The reviewer may have increased, but maybe no more than 5.

qzrK gave a 6. The main weaknesses were lack of rigor and lack of SOTA methods in the experimental setup. Authors have provided feedback related to these two points, but in my opinion it is sufficient to ensure that the reviewer keeps his score, but I am not sure that he would have increased his score.

---

### Decision · Program_Chairs · 2026-01-26

Accept (Poster)